# Structural insights into Siglec-15 reveal glycosylation dependency for its interaction with T cells through integrin CD11b

Maria Pia Lenza[1,11], Leire Egia-Mendikute[2,11], Asier Antoñana-Vildosola[2,11], Cátia O. Soares[3,4], Helena Coelho[3,4], Francisco Corzana[5], Alexandre Bosch[2], Prodhi Manisha[2], Jon Imanol Quintana[1], Iker Oyenarte[1], Luca Unione[1,6], María Jesús Moure[1], Mikel Azkargorta[7], Unai Atxabal[1], Klaudia Sobczak[1], Felix Elortza[7], James D. Sutherland[8], Rosa Barrio[8], Filipa Marcelo[3,4], Jesús Jiménez-Barbero[1,6,9,10] ✉, Asis Palazon[2,6] ✉ & June Ereño-Orbea[1,6] ✉

Sialic acid-binding Ig-like lectin 15 (Siglec-15) is an immune modulator and emerging cancer immunotherapy target. However, limited understanding of its structure and mechanism of action restrains the development of drug candidates that unleash its full therapeutic potential. In this study, we elucidate the crystal structure of Siglec-15 and its binding epitope via co-crystallization with an anti-Siglec-15 blocking antibody. Using saturation transfer-difference nuclear magnetic resonance (STD-NMR) spectroscopy and molecular dynamics simulations, we reveal Siglec-15 binding mode to α(2,3)- and α(2,6)-linked sialic acids and the cancer-associated sialyl-Tn (STn) glycoform. We demonstrate that binding of Siglec-15 to T cells, which lack STn expression, depends on the presence of α(2,3)- and α(2,6)-linked sialoglycans. Furthermore, we identify the leukocyte integrin CD11b as a Siglec-15 binding partner on human T cells. Collectively, our findings provide an integrated understanding of the structural features of Siglec-15 and emphasize glycosylation as a crucial factor in controlling T cell responses.

Tumor evasion mechanisms that suppress the cytotoxic activity of tumor infiltrating lymphocytes (TILs) are a major obstacle for cancer immunotherapy[1–5]. The identification of immune checkpoint receptors −such as programmed cell death-1 (PD-1) and cytotoxic T lymphocyte antigen 4 (CTLA-4)−and their counterpart ligands paved the way for the development of more efficacious immunotherapies based on immune checkpoint blockade (ICB)[6–9]. However, despite the marked clinical success of ICB and their subsequent regulatory approval for the treatment of several types of tumors, many patients do not respond, relapse, or are not eligible for current treatments.

Protein glycosylation is a post-translational modification that governs a wide variety of cellular processes in health and disease[10,11].

Aberrant glycosylation in cancer cells has been shown to contribute to tumor progression and metastatic potential[12–15]. One of the most prevalent glycan alterations found in tumor cells is sialyl-Tn (STn) (Neu5Acα2,6-GalNAcα1-O-Ser/Thr), which results from dysfunctional T synthase and/or aberrant N-acetylgalactosamine (GalNAc)-transferase activity[16–19].

Sialic acid-binding immunoglobulin (Ig)-like lectin 15 (Siglec-15) is a single-pass transmembrane protein initially described in osteoclasts[20,21]. Siglec-15 presents an extracellular domain containing a conserved N-terminal variable (V)-set Ig domain, which binds sialic acid, and a constant 2 (C2)-set Ig domain[22]. This V-set domain folds into a sandwich of two β-pleated sheets consisting of antiparallel β-strands and differs from C2-set by having additional β-strands within the β-

sheets[23,24]. Siglec-15 triggers a signaling cascade through its positively charged transmembrane region after association with the immunoreceptor tyrosine-based activation motif (ITAM) adapter proteins DAP10 or DAP12, regulating several biological processes including osteoclast maturation, bone remodeling and susceptibility to fungal infections[25–28]. Siglec-15 has recently emerged as a modulator of immune responses that can be expressed by tumor-associated macrophages (TAMs)[29]. Ligation of Siglec-15 suppresses antigen-specific T cell responses, and monoclonal antibodies (mAbs) that block the interaction of Siglec-15 to its binding partner(s) promote antitumor responses and are under evaluation for the treatment of several types of cancer[29–32].

The preferential glycan partners of Siglec-15 remain controversial. Initial studies identified an interaction between STn and Siglec-15[21]. However, more recent studies showed that Siglec-15 can bind branched α(2,3) and α(2,6) di-sialylated biantennary and triantennary N-glycans[33], exhibiting high-avidity ligation to (2,3)- and (2,6)-bound sialic acids compared to STn in the context of high-affinity synthetic sialic acid analogs[34]. Moreover, Siglec-15 shows robust binding to sulfated sialic acid containing glycans[35,36]. Binding constants of Siglecs for the N-acetylneuraminic acid (Neu5Ac) linked by α(2,3)- or α(2,6)- mono- or di-saccharides are in the low millimolar range (Kd of 0.1–3 mM)[37,38].

In this work, we have determined the crystal structure of Siglec-15, bound to a blocking mAb, at 2.1 Å resolution. The synergistic combination of X-ray crystallography, nuclear magnetic resonance (NMR) spectroscopy and molecular modeling methodologies have allowed us to delineate the sialic acid-binding pocket of Siglec-15 and provided a comprehensive characterization of its interaction dynamics with α(2,3)- and α(2,6)- sialyllactose and STn-Ser antigen. Binding assays performed with human T cells, which lack detectable expression of STn, have revealed that α(2,3)- and α(2,6)-linked sialic acids are required for Siglec-15 ligation. We have also identified CD11b, a member of the leukocyte-restricted β2 integrin family, as a binding partner for Siglec-15. Biochemical, biophysical, and functional studies have unequivocally indicated a sialylation dependency for the interaction of Siglec-15 with CD11b. Together, our results provide structural insights into the carbohydrate recognition domain of Siglec-15 and suggest that glycosylation may regulate receptor-binding interactions that result in T cell suppression.

## Results

### Structural elucidation of Siglec-15 determined by co-crystallization with an anti-Siglec-15 mAb

To assist the crystallization of the extracellular domain (ECD) of Siglec-15, that consists of two Ig domains (d1 and d2) (Siglec-15$_{d1-d2}$), we employed the fragment antigen-binding (Fab) of an anti-Siglec-15 mAb as a crystallization chaperone (clone 5G12[32]). This non-glycosylated Fab was produced by cloning its variable heavy (VH) and light chains (VL) into a Fab scaffold containing human constant heavy (CH) and kappa light chains (CL). First, binding of 5G12 Fab to Siglec-15$_{d1-d2}$ was analyzed by biolayer interferometry (BLI) (Supplementary Fig. 1). In agreement with the previous characterization of this antibody[32], the affinity of the 5G12 Fab is in the low nanomolar range ($K_D = 4.68 \pm 0.29$ nM) (Supplementary Fig. 1). The X-ray crystallography 3D structure was determined by molecular replacement at 2.1 Å resolution in C121 space group ($a = 216.53$, $b = 60.53$ and $c = 53.37$ Å; $\alpha = 90.00$, $\beta = 100.82$, and $\gamma = 90.00°$), using the crystal structure of the 5G12 Fab (at 3.9 Å resolution) as initial search model (Supplementary Table 1). The asymmetric unit contains one molecule of Siglec-15 and one molecule of 5G12 Fab. The electron density map allowed us to manually build the V-set domain of Siglec-15 (Supplementary Fig. 2a). Further analysis of the molecular weight by SDS-PAGE and nano-scale liquid chromatographic tandem mass spectrometry (nLC–MS/MS) for the identification of peptides present on the crystal confirmed the

presence of both domains (d1 and d2) of Siglec-15 (Supplementary Fig. 2b). However, the required electron density to build d2 on Siglec-15 was missing, likely due to its intrinsic high flexibility.

The V-set domain of Siglec-15 is composed of two β-sheets made of β-strands AA´BB´ED and CC´C´´FGG´, which are connected by a C64-C142 disulfide linkage (Fig. 1a–c). As expected, the electron density that justifies the presence of any N- or O-linked glycans on the surface of Siglec-15 was not observed. The predicted canonical ligand-binding pocket contains the key R143 residue at strand F, which serves to generate the conserved salt bridge with the negatively charged carboxylate C1 of sialic acid[39]. Interestingly, the V-set domain of Siglec-15 contains an extra β-strand, hereafter called C´´. Moreover, the C´–C´´ loop connects to the C´ strand with the C95-C104 disulfide bridge. This unique feature generates an extended surface area (of 6053 Å$^2$) on the CC´C´´FG face of the Ig domain in Siglec-15 (Supplementary Fig. 3). Interestingly, the structural superposition of the carbohydrate recognition domain with other members of the Siglec family showed that Siglec-15 is similar to Siglec-1 (Sialoadhesin) (r.m.s.d. 0.723) (Supplementary Fig. 3).

The analysis of the crystal structure of the Siglec-15-5G12 complex showed that Fab clone 5G12 binds to the carbohydrate recognition domain on Siglec-15. Indeed, 5G12 binds primarily at the interface between the two β-sheets on the V domain (Fig. 1a). The 5G12 binding epitope consists of 890 Å$^2$ of buried surface area (BSA) (Supplementary Table 2). In particular, the 5G12 heavy-chain complementarity determining regions 1 and 3 (HCDR1 and HCRD3) interact with the C–C´ and G-G´ loops and the F strand of Siglec-15 (Fig. 1b). Additionally, the light-chain CDR3 (LCDR3) makes polar contacts with the G strand and G–G´ loop (Fig. 1b). The superimposition of the unliganded and Siglec-15-liganded 5G12 Fab indicates that its paratope is pre-organized for efficiently binding its antigenic site (r.m.s.d. of 0.57 Å) (Supplementary Fig. 4).

### Siglec-15 binds to α(2,3) and α2,6-linked sialoglycans on human T cells, which do not express STn

Based on the finding that Siglec-15 binds to STn[21], and aiming to unravel the potential binding partners of Siglec-15 on T cells, the expression of STn on human T cells was assessed. No STn expression was detected in human T cells, irrespective of their activation status, while cell models of leukemia express significant levels of STn on their cell surface (Fig. 2a). The presence of other possible sialic acid containing glycans (sialoglycans) on T cells that could act as potential binding partners for Siglec-15 was then evaluated. We focused on α(2,3) or α(2,6)-linked sialoglycans by measuring the binding capacity of two well-characterized lectins, *Maackia amurensis* agglutinin (MAL II, which specifically recognizes α(2,3)-linked sialic acids) and *Sambucus nigra* agglutinin (SNA, which specifically recognizes α2,6-linked sialic acids), to T cells. The obtained data unambiguously demonstrated that both α(2,3) and α(2,6)-linked sialoglycans are present on the surface of human T cells (Fig. 2b, c). Furthermore, when activated T cells were preincubated with Siglec-15, the binding ability of SNA and MAL II to T cells markedly decreased (Fig. 2d), supporting that Siglec-15 recognizes both the α(2,3) and α(2,6) sialoglycans present on T cells.

To confirm the glycosylation dependency of the binding of Siglec-15 to human T cells, T cells were pre-treated with a pan-deglycosylation enzymatic cocktail. Fittingly, removal of all N-linked glycans and many common O-linked glycans on T cells abrogated the binding of Siglec-15 (Fig. 2e). Focusing on the relevance of sialylated glycans present on T cells, sialic acid moieties on α(2,3) and α(2,6) sialoglycans were specifically removed from the surface of T cells through the action of neuraminidase A (Fig. 2f), or α(2,3) sialoglycans with neuraminidase S (Fig. 2g). The observed decrease on Siglec-15 binding upon neuraminidase A desialylation was comparable to that measured in the case of deglycosylation (Fig. 2e), indicating that the sialic acid moiety is the key unit for the binding of Siglec-15 to human T cells. This observation

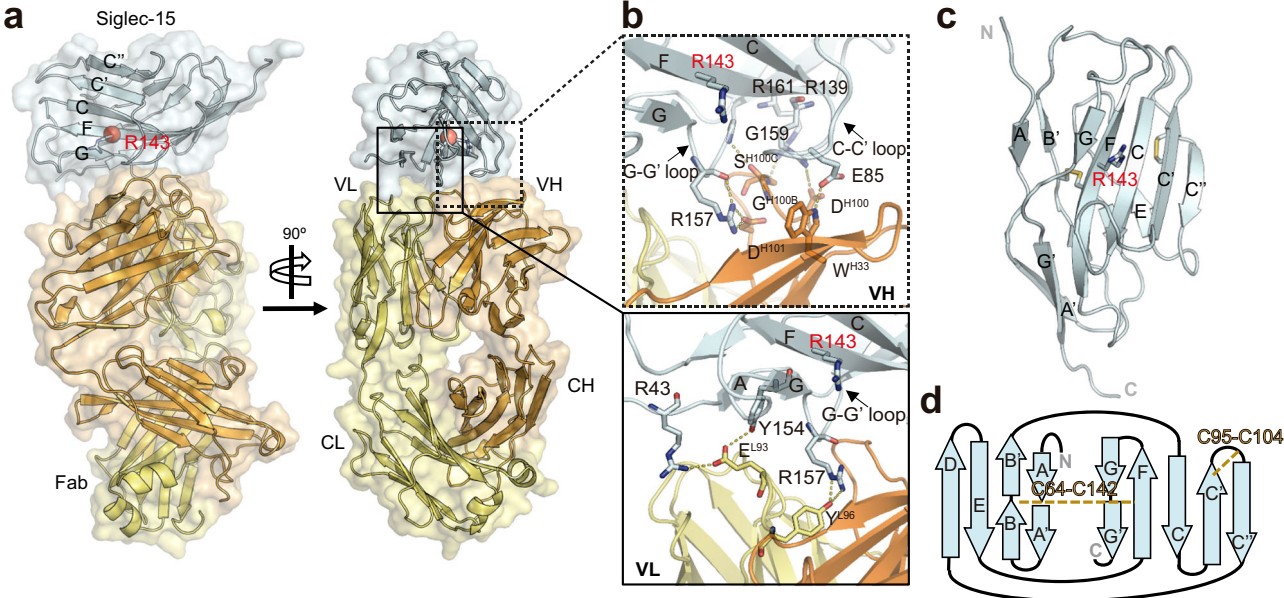

**Fig. 1 | Overall structure of Siglec-15 in complex with anti-Siglec-15 Fab.**
**a** Cartoon and surface representation of two views of the crystal structure of Siglec-15 in complex with anti-Siglec-15 5G12 Fab. 5G12 is composed of a heavy chain (HC) (in orange) and a light chain (LC) (in yellow). The variable (VH and VL) region of 5G12 Fab binds to the V-set domain of Siglec-15 (in cyan). The conserved R143 residue on Siglec-15 ligand-binding site that forms the salt bridge with the carboxylate C1 of sialic acid moiety is represented with a red sphere. **b** Zoom-in views of the interaction between Siglec-15 and VH (top) and VL (bottom) domains of 5G12 Fab. The heavy-chain CDRs 1 and 3 (HCDR1 and HCRD3) (in orange) of 5G12 interact with C−C′ and G-G′ loops and the F strand of Siglec-15 (cyan). The light-chain CDR3 (LCDR3) from 5G12 (in yellow) interacts with the G strand and G-G´ loop (in blue). Polar contacts (H-bonds and salt bridges) are represented with yellow dashed lines. **c** Cartoon representation of the two β-sheets forming the V-set Ig domain of Siglec-15. The internal disulfide linkages (between C64 and C142; and between C95 and C104) are represented as sticks. **d** Topology diagram of V type Ig-like domain in Siglec-15. The β-sheets of are formed by strands AA´BED and C″C′CFGG′. C64-C142 and C95-C104 intra-disulfide linkages are represented with dashed yellow lines.

was further supported by an additional assay that compared the binding capacity of the wild type (WT) vs the key R143A mutant of recombinant Siglec-15 towards human T cells. The substitution of the conserved R143 in the binding pocket to Ala143 abolishes the binding capacity of Siglec-15 towards sialic acids[21,40]. Indeed, the R143A mutant Siglec-15, as opposed to WT Siglec-15, is unable to bind human CD8[+] and CD4[+] T cells (Fig. 2h).

### The antibody-mediated blockade of Siglec-15 interferes with the sialic acid-binding site

To evaluate the ability of 5G12 Fab to impede the interaction between sialoglycans present on CD4[+] and CD8[+] T cells and Siglec-15, blocking assays were carried out, and monitored by flow cytometry. The data showed that Siglec-15 loses its ability to bind to CD4[+] and CD8[+] activated human T cells in the presence of 5G12 anti-Siglec-15 Fab (Fig. 3a, b).

Complementarily, STD-NMR-based competition binding experiments were carried out with α(2,3)-, α(2,6)-sialyllactose (3′SL and 6′SL) derivatives (Fig. 3c) and STn-Ser (Supplementary Fig. 5) ligands and recombinant Siglec-15 in the absence and presence of the 5G12 Fab. The STD-NMR responses corresponding to all interrogated ligands, including the NHAc methyl group of Neu5Ac, were highly diminished by the presence of 5G12 Fab. These results clearly indicate that the interaction with the 5G12 Fab precludes the binding of sialoglycans to Siglec-15, strongly suggesting that Siglec-15 interacts with T cells through its sialic acid recognition domain.

### Molecular basis of α(2,3)-, α(2,6)-sialyllactose and STn binding to Siglec-15 by NMR and molecular modeling

The analysis of the STD-NMR experiments carried out on the complexes of 3′SL, 6′SL or STn-Ser with Siglec-15, allowed the detailed description of their interactions at the molecular level (Fig. 4a and Supplementary Fig. 6). As expected, the STD-NMR-derived epitope map highlighted the relevance of the *N*-acetylneuraminic acid

(Neu5Ac) in the binding event, as previously observed for other Siglecs[41]. For STn-Ser, the binding preference towards the Neu5Ac moiety was particularly evident by comparing the STD response of the two NHAc-Neu5Ac and NHAc-GalNAc methyl groups. The requirement of the presence of Neu5Ac for the interaction with Siglec-15 was further confirmed by the lack of STD-NMR response when the non-sialylated lactose and Tn-Ser fragments were used (Supplementary Fig. 7).

The Siglec-15 bound conformation of these ligands was also investigated by NOESY NMR experiments (Supplementary Figs. 8, 9 and 10), allowing to deduce their bound conformations (Fig. 4b). For 3′SL, the detailed inspection of the NOESY spectra (free and bound state) evidenced the existence of a conformational equilibrium in the free state around the φ torsion of the α(2,3) Neu5Ac-Gal linkage. However, upon Siglec-15 binding, a conformational selection process took place, with the exclusive presence of the -g conformer (Supplementary Fig. 8). For the 6′SL analog, which is also rather flexible in free solution, a major -g geometry around the α(2,6) Neu5Ac-Gal linkage (Supplementary Fig. 9) could also be deduced for the bound state, while the conformation around ω angle, which is dominated by the gt rotamer in the free state, could not be deduced in a non-ambiguous manner due to overlapping of the key NOE cross peaks (Supplementary Fig. 9). A similar conformational behavior was inferred for STn-Ser (Supplementary Fig. 10).

3D model structures of the Siglec-15/sialoglycans complexes were further obtained by molecular dynamics (MD) simulations as described in the Methods section. Fittingly, all glycan-lectin complexes were stable during the whole MD simulation and the Neu5Ac moiety was found to establish key stabilizing contacts with the protein for all the sialoglycans (Fig. 4, Supplementary Figs. 11, 12, 13 and 14), in full agreement with the STD-derived epitope map.

The analysis of the models of the complexes of Siglec-15 bound to 3′SL, 6′SL and STn-Ser allowed explaining why Fab 5G12 efficiently

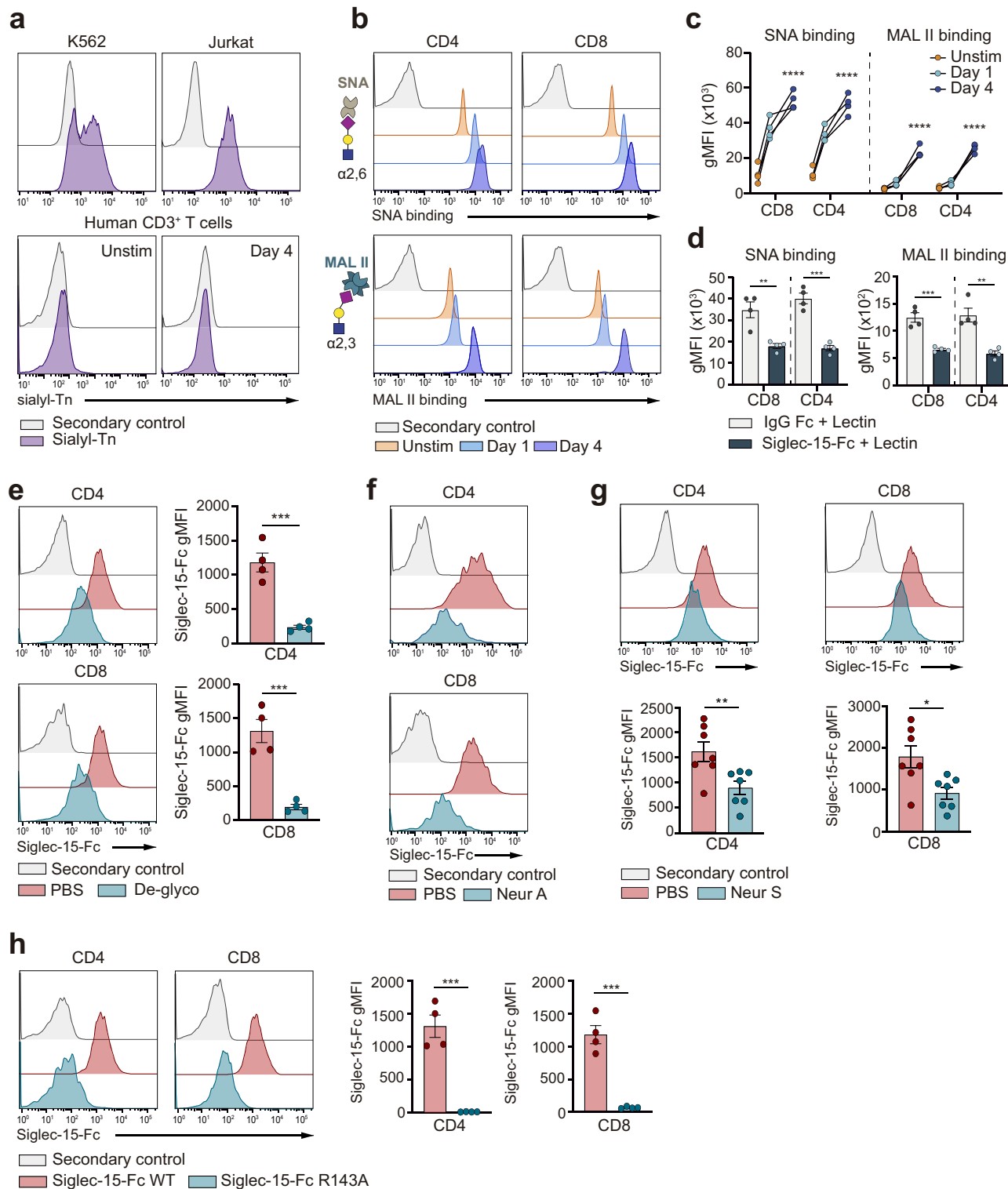

blocks the binding of the sialoglycans (Supplementary Fig. 15), since its interaction takes places at the same locus. It also supports the absence of sialylated glycans (e.g. STn-Ser) in the crystal of the Siglec-15-5G12 complex after co-crystallization and soaking attempts with these ligands.

### Siglec-15 binds to CD11b on human T cells via sialic acid

Next, a proximity labeling assay based on tyramide radicalization coupled with proteomics[42] was performed to identify the putative sialic acid containing glycoproteins that act as binding partners of Siglec-15 on the surface of T cells. Several cell membrane-associated glycoprotein candidates were identified by using mass spectrometry, including highly glycosylated mucins and several components of the TCR and immune synapse (e.g., HLA-I, TCR β-chain, CD44) (Supplementary Table 3). Interestingly, both CD11b and CD18 integrins, which form a heterodimer, were also identified. Given that other members of the Siglec family can regulate CD11b signaling on a sialylation-dependent mechanism[43], the interaction of Siglec-15 with CD11b was explored. In particular, ELISA (Fig. 5a) and co-immunoprecipitation (Fig. 5b) assays were carried out to investigate whether Siglec-15 can

**Fig. 2 | Binding of Siglec-15 to human T cells depends on α(2,3) and α(2,6) sialylation. a** Representative flow histograms showing the expression of STn on leukemia cell lines (K562 and Jurkat), compared to unstimulated (unstim) or activated human T cells. **b** Representative histograms of SNA and MAL II lectin binding to the surface of unstimulated or activated T cells. The general glycan structure recognized by SNA and MAL II lectins is drawed using the Symbol Nomenclature for Glycans (SNFG). Sialic acid with magenta rhomboid, galactose (Gal) with yellow circle and N-acetylglucosamine (GalNAc) with blue square. **c** The binding of lectins (SNA or MAL II) to CD8+ or CD4+ T cells before and after activation was quantified by flow cytometry (SNA CD8+: $p < 0.0001$, SNA CD4+: $p < 0.0001$, MAL II CD8+: $p < 0.0001$, MAL II CD4+: $p < 0.0001$, $n = 4$ donors). **d** Bar graphs representing the binding of SNA and MAL II lectins to activated T cells after preincubation with Siglec-15 or IgG-Fc control (SNA CD8+: $p = 0.0047$, SNA CD4+: $p = 0.0002$, MAL II CD8+: $p = 0.0006$, MAL II CD4+: $p = 0.002$, $n = 4$ donors). **e** Representative flow

cytometric histograms (left) and pooled data (right) of Siglec-15-Fc binding after pan-deglycosylation (De-glyco) treatment of human CD8+ and CD4+ T cells (CD4+: $p = 0.0005$, CD8+: $0.0007$, $n = 4$ donors). **f** Representative histograms of Siglec-15-Fc binding after desialylation with Neuraminidase A (Neur A) of human CD8+ and CD4+ T cells measured by flow cytometry ($n = 4$ donors). **g** Representative flow cytometric histograms (left) and pooled data (right) showing the binding of Siglec-15-Fc to human CD8+ and CD4+ T cells treated with Neuraminidase S (Neur S) (CD4+: $0.007$, CD8+: $0.011$, $n = 7$ donors). **h** Representative histograms (top) and pooled data (bottom) of Siglec-15-Fc R143A binding to activated CD8+ and CD4+ T cells (CD4+: $0.0002$, CD8+: $0.0003$, $n = 4$ healthy donors). Secondary control means that only an anti-Fc detector antibody, but not recombinant Fc-chimera protein was added to the sample. Error bars denote SEM. *$p < 0.05$, **$p < 0.01$; ***$p < 0.001$; ****$p < 0.0001$) as determined by two-tailed, unpaired Student's t test. Source data are provided as a Source Data file.

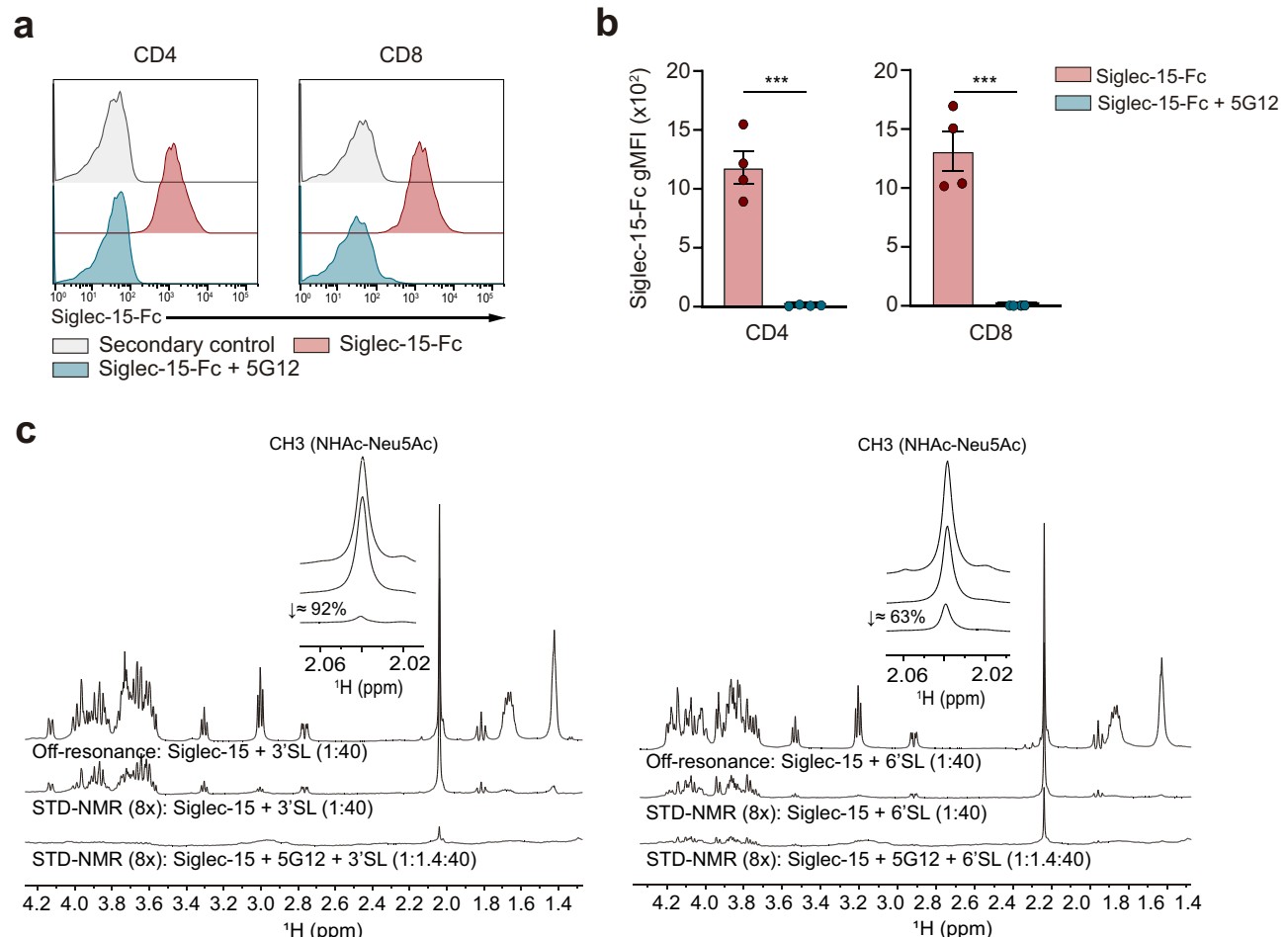

**Fig. 3 | Anti-Siglec-15 blocking mAb 5G12 competes for the sialic acid-binding site of Siglec-15. a** Representative flow cytometry histograms of activated CD8+ and CD4+ human T cells showing the binding of recombinant Siglec-15-Fc in the presence or absence of anti-Siglec-15 5G12 Fab. Here, secondary control means that only an anti-Fc detector antibody, but not recombinant Fc-chimera protein was

added to the sample. **b** Bar graphs show Siglec-15-Fc binding to human T cells in the presence or absence of 5G12 Fab (CD4+: $0.0001$, CD8+: $0.0002$, $n = 4$ donors). Errors bars denote SEM. ***$p < 0.001$ as determined by two-tailed, unpaired Student's $t$ test. **c** Competition of 3'SL (left)/6'SL (right) and 5G12 mAb for the same binding site of Siglec-15. Source data are provided as a Source Data file.

directly bind to CD11b. Fittingly, the existence of interaction between CD11b and Siglec-15 WT was confirmed. In contrast, the capacity of Siglec-15 R143A mutant to bind to CD11b was dramatically reduced, demonstrating the requirement of a functional sialic acid-binding domain in Siglec-15 for the interaction with CD11b (Fig. 5a, b). Furthermore, this direct interaction was confirmed by STD-NMR competition binding experiments, which showed that the STD-NMR signal intensities corresponding to the sialic acid protons markedly

decreased upon addition of the CD11b/CD18 heterodimer (Fig. 5c and Supplementary Fig. 16). To study the relevance of this interaction in human T cells, the presence of CD11b on human T cells was first examined to confirm its robust surface expression in activated CD4+ and CD8+ T cells (Fig. 5d and Supplementary Fig. 17a). Subsequent blocking assays explored the binding of Siglec-15 to T cells in the presence of a blocking anti-CD11b antibody (clone M1/70)[44]. The obtained results showed that the blockade of CD11b with clone M1/

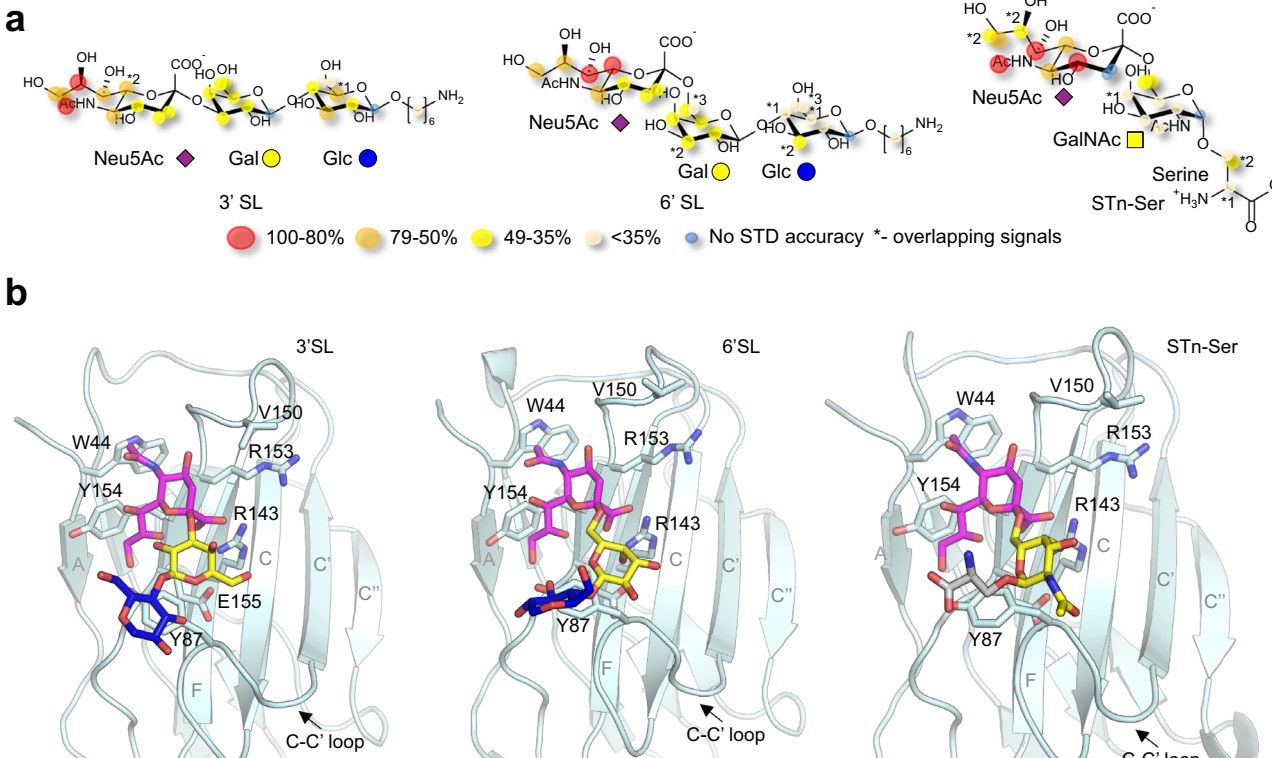

**Fig. 4 | Molecular recognition of 3'SL, 6'SL and STn-Ser by Siglec-15. a** STD-based epitope mapping for 3'SL, 6'SL and STn-Ser in the presence of Siglec-15. The relative STD response is coded according to the legend. **b** Representative frames derived from 0.5 μs molecular dynamics (MD) simulations of 3'SL (left), 6'SL (middle) and STn-Ser (right) (sticks, purple for Neu5Ac, yellow for GalNAc and Gal, blue for Glc, and gray for serine) in complex with Siglec-15 V-set domain (cartoon and sticks, light cyan). The residues of the protein involved in interactions with the ligands are identified and represented in sticks. Besides the essential salt bridge between Neu5Ac carboxylate group and R143 guanidinium group, four hydrogen bonds are maintained throughout the simulation: OH9-Neu5Ac with CO-E155; O9-Neu5Ac with OH-Y87; OH4-Neu5Ac with CO-V150; and NHAc-Neu5Ac with CO-R153. In addition, CH−π interactions are established between C9-Neu5Ac and Y154; NHAc-Neu5Ac and W44. Source data are provided as a Source Data file.

70 significantly reduces the binding of Siglec-15 to human T cells (Fig. 5e), as opposed to the anti-CD11b antibody clone CBRM1/5, which binds to the active conformation of I domain on CD11b[45] (Supplementary Fig. 17b).

Additional assays to demonstrate the binding of Siglec-15 to CD11b expressed on human CD4[+] and CD8[+]T cells were then carried out. The knockdown of CD11b resulted in reduced binding of Siglec-15 (Fig. 5f and Supplementary Fig. 17c). On the other hand, over-expression of CD11b/CD18 increased the binding of Siglec-15 to T cells (Fig. 5g and Supplementary Fig. 17d). In light of these findings, we decided to explore the sialylation pattern of CD11b in human T cells. To this end, protein extracts of activated human T cells were treated with different glycosidases. Interestingly, the treatment with N-glycosidases, but not with O-glycosidases or neuraminidase A, resulted in a dramatic change in the electrophoretic mobility (Fig. 6a).

Additionally, the presence of α(2,3) and α(2,6) sialoglycans displayed on CD11b expressed on T cells was further analyzed. The western blot of purified CD11b from T cells showed that SNA binds to CD11b, but not MAL II, suggesting that CD11b contains α(2,6) sialoglycans (Fig. 6b). These experimental findings were employed to build a putative model of the interaction of Siglec-15 and the sialylated N-glycans at CD11b/CD18 (Fig. 6c).

## Discussion

Although the development of immune checkpoint inhibitors (ICIs) has achieved significant clinical benefit in the treatment of cancer, an important medical need still remains[46]. In this context, the discovery of Siglec-15 as an immune suppressor has raised the interest for this molecule from a therapeutic perspective[29], especially since its expression profile differs from PD-L1. Here, we provide insights into the structure of Siglec-15 and its molecular recognition features when interacting with its sialylated partners. The analysis of the crystal structure of Siglec-15 has shown a peculiarity in the V-set domain in this family of lectins. Siglec-15, unlike other Siglecs[47], displays an extra β-strand (C´´) that makes a larger accessible interacting surface for ligands in Siglec-15.

Our findings demonstrate that the presence of a well-defined glycan recognition site is essential for the binding of Siglec-15 to T cells. Moreover, the presence of α(2,3) and α(2,6) sialylated glycans in T cells as the main binders for Siglec-15 has also been shown. The experiments performed with a mutant version of Siglec-15 lacking the sialic acid-binding capacity have further demonstrated that the interaction of Siglec-15 with T cells depends on sialylation. The combined NMR and X-Ray crystallography studies have allowed assessing that the 5G12 Fab portion blocks the interaction of Siglec-15 with sialylated ligands. This structural evidence is also aligned with the binding features of recombinant Siglec-15 to human CD4[+] and CD8[+] T cells in the absence and presence of the 5G12 Fab. Moreover, glycosylation on human T cells, particularly sialylation, is essential for the binding of Siglec-15.

Human T cells, as opposed to the Jurkat cell line that contains a loss-of-function mutation in *Cosmc*[48], do not express Sialyl-Tn antigen (STn). This fact prompted us to explore the presence of other Siglec-15 glycan binders on human primary T cells. Indeed, α(2,3) and α(2,6) sialylated glycans are present in T cells and are the main acceptors for Siglec-15. Experiments performed with a mutant version of Siglec-15

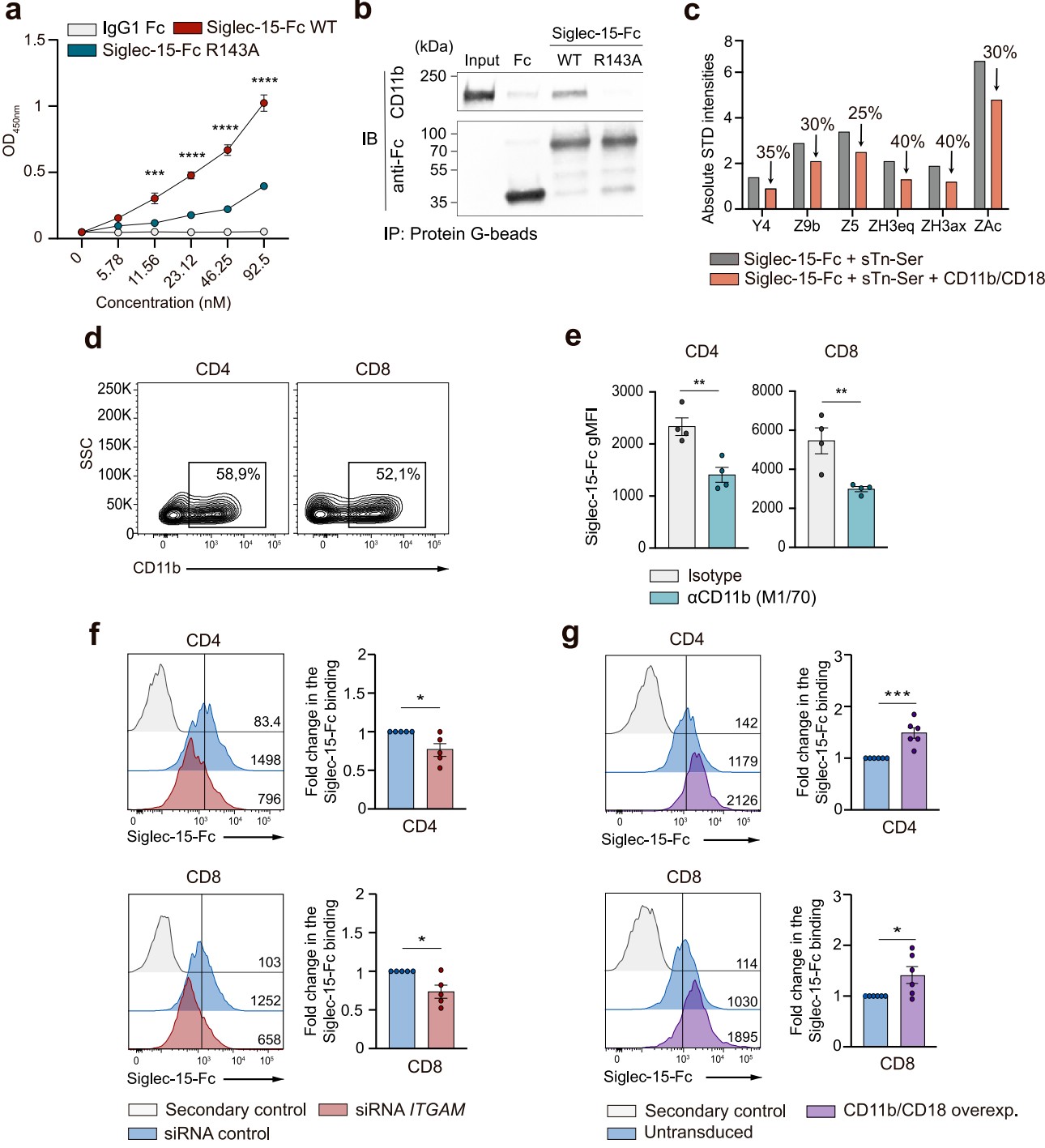

**Fig. 5 | Siglec-15 binds to CD11b in a sialic acid-dependent manner. a** OD values corresponding to an ELISA performed with serially diluted IgG1-Fc (gray), Siglec-15-Fc WT (red) or Siglec-15-Fc R143A (blue) against plate-coated CD11b/CD18. Averages of triplicates are shown. **b** Co-immunoprecipitation showing the interaction of Siglec-15-Fc or Siglec-15-Fc R143A with CD11b. Assay repeated twice with similar results. **c** Bar graph representing the absolute STD-NMR intensities corresponding to proton signals of STn-Ser + Siglec-15 before the addition of CD11b/CD18 (gray), and after (orange). **d** Representative contour plots showing the expression of CD11b on activated CD4+ and CD8+ T cells by flow cytometry. **e** Quantitation of Siglec-15-Fc binding to CD4+ and CD8+ T cells in the presence or absence of anti-CD11b blocking mAb (clone M1/70), measured by flow cytometry (CD4+: $p = 0.0052$, CD8+: 0.0098, $n = 4$ donors). **f** Representative flow cytometric histograms (gMFI) and pooled data showing the fold change in the binding of Siglec-15–Fc to T cells transfected with indicated siRNAs (CD4+: $p = 0.046$, CD8+: $p = 0.036$, $n = 5$ donors). **g** Representative flow cytometric histograms (gMFI) and pooled data of the fold change in the binding of Siglec-15-Fc to T cells over-expressing CD11b/CD18 compared to untransduced cells (CD4+: 0.0007, CD8+: 0.036, $n = 6$ donors). Error bars denote SEM. *$p < 0.05$, **$p < 0.01$; ***$p < 0.001$; ****$p < 0.0001$) as determined by two-tailed, unpaired Student's $t$ test. Source data are provided as a Source Data file.

lacking the sialic acid-binding capacity further demonstrated that the interaction of Siglec-15 with T cells depends on sialylation.

The role of glycosylation on cancer progression is well characterized[14] and the impact of sialylation on the modulation of immunity in health and disease is also widely accepted[22]. Moreover, the family of Siglecs influences innate and adaptive immune responses in cancer[49,50]. As a result, many emerging therapeutic agents are directly targeting glycosylation or glycan-based pathways[11,51]. They include

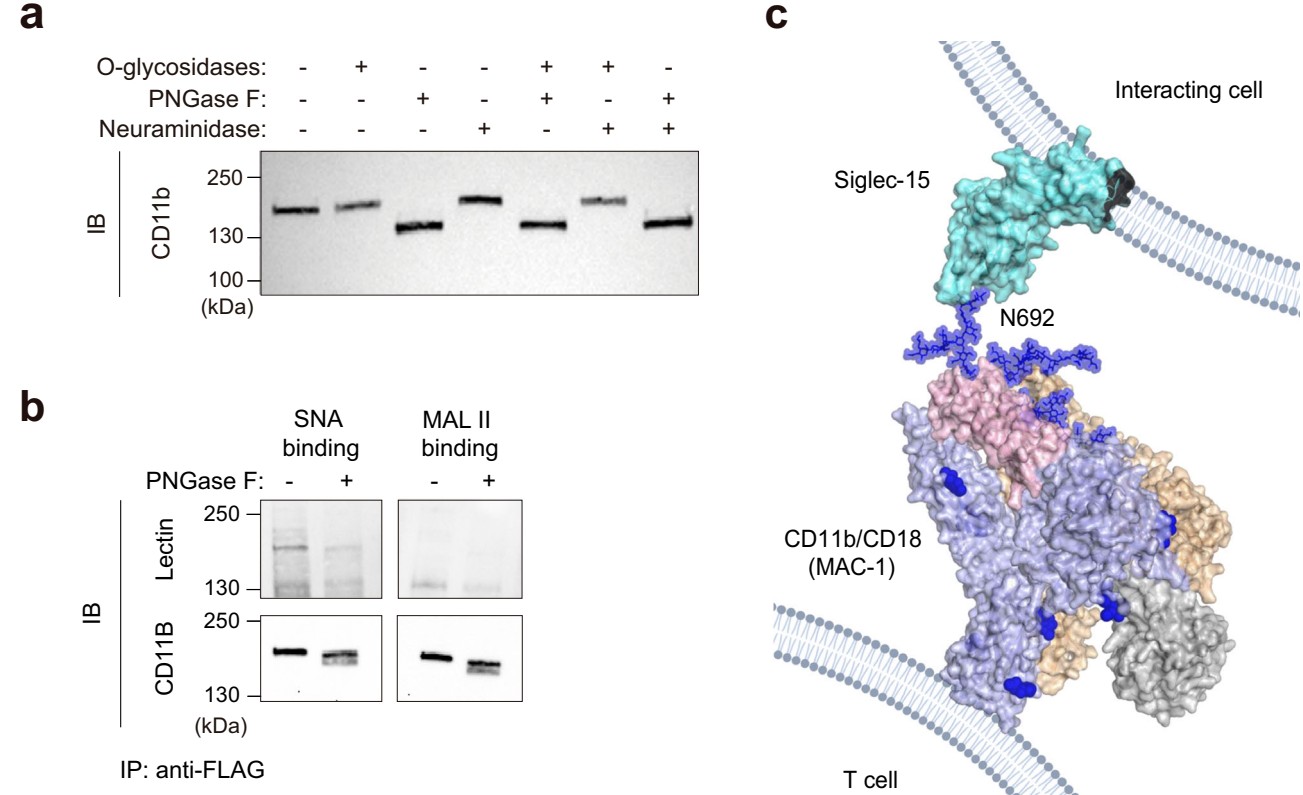

**Fig. 6 | Glycosylation pattern of CD11b expressed on T cells. a** Western blot of CD11b on T cell lysates treated with PNGase F, O-glycosidases or Neuraminidase A for 4 h as indicated. **b** Lectin blot analysis of purified CD11b-flag from transduced human CD3[+] T cells. Complete membrane blots are included in Supplementary Fig. 18. **a, b** One representative western blot from *n* = 2 biologically independent experiments. **c** Model of the interaction between Siglec-15 present on the surface of interacting cells and CD11b/CD18 integrin from the cell surface of T cells. The full extracellular domain of Siglec-15 (cyan) was manually built using the crystal structure of the V-set domain and the C2-type Ig-like domain of CD22 (PDB ID:5VKJ). The full extracellular domain of CD11b (blue)/CD18 (wheat) (PDB ID 7USM)[73] heterodimer (from the cell surface of T cells) is represented. The I domain (taken from PDB ID 3K72) and the thigh region of CD11b are colored in gray and pink, respectively. The N-linked glycans present in CD11b are represented as blue spheres. In this model, Siglec-15 binding pocket at V domain is interacting with the sialylated N692-linked glycan of CD11b.

classic immunotherapy approaches based on mAbs or cell therapy[52,53], or more sophisticated strategies, such as those based on targeted glycan degradation[54]. The identification of novel sialylation-dependent interactions and glyco-immune checkpoint receptors is an area of active research with relevant implications for designing the next generation of immunotherapies.

Herein, we have used an established proximity labeling assay[42] to identify several Siglec-15 glycosylated binders, including hits that are known to locate at the site of the immunological synapse, suggesting that Siglec-15 might interfere with this process to modulate T cell activity. One of the best known integrins that regulates T cell activation is CD18, which can form heterodimers with LFA-1 or CD11b, among others. The regulation of T cell effector function by the LFA-1/CD18 heterodimer has been recently described[55], although the function of CD11b/CD18 in T cells remains largely unexplored. As shown in this work, the interaction of Siglec-15 with human T cells depends on their level of CD11b expression. However, we can not exclude that other glycoproteins may also contribute to the binding of Siglec-15 to T cells.

The immunoprecipitation and ELISA assays have unambiguously indicated that the binding of Siglec-15 to CD11b takes place through its V domain and that this interaction is abolished by introducing the point mutation R143A residue. We have also demonstrated that Siglec-15 recognizes the sialic acids present on CD11b/CD18, as deduced from CD11b enriched protein extracts of activated human T cells. Additionally, Siglec-15 binding in the presence of an anti-CD11b blocking mAb (clone M1/70) was reduced. The binding epitope for this M1/70

clone is located between residues 614-682 on CD11b[56], in conformational proximity to its N692 and N696 N-glycosylation sites, which we propose to take part in the interaction with Siglec-15 (Fig. 6c).

In summary, the structural and molecular recognition features of Siglec-15 have been unraveled, along with the relevance of the glycosylation pattern on T cells for Siglec-15 binding. Moreover, the CD11b/CD18 heterodimer on T cells has been identified as a natural binder for Siglec-15. All of these data support that glycosylation regulates the receptor-binding interactions that result in T cell suppression.

## Methods

### Cell lines
HEK293F (R79007, Thermo Fisher) and HEK293S (CRL-3022) cells were grown at 37 °C, 70% humidity 8% $CO_2$ at 130 rpm in Freestyle media (12338018, Thermo Fisher). Jurkat cells (TIB-152, ATCC) and K562 cells (CCL-234, ATCC) cells were grown at 37 °C in RPMI with 10 % FBS and penicillin/streptomycin (15140122, Gibco). The Lenti-X 293T cell line (632180, Takara Bio Inc.) was grown in DMEM (41966-029, Gibco) with 10 % FBS and penicillin/streptomycin.

### Primary human T cells
T cells were obtained from buffy coats of healthy donors (Biobanco Vasco, BIOEF) after ethical approval (PI + CES-BIOEF 2019-08). Briefly, PBMCs were separated by gradient differentiation using Ficoll-Histopaque (17-1440-03, Fisher scientific), and CD3[+] T cells were purified by negative selection using EasySep™ Human T Cell

Enrichment Kit (Stemcell) following manufacturer's instructions. Purity was confirmed by flow cytometry to be >95%. T cells were then activated with anti-CD3/CD28 Dynabeads (11131D, Thermo Fisher) in CST OpTimizer medium (A1048501, Gibco) supplemented with IL-2 at 100 IU/mL (130-097-743, Miltenyi Biotec).

## Construct design of Siglec-15 extracellular domain and 5G12 Fab proteins

The DNA encoding the full-length extracellular domain (d1-d2) of human Siglec-15 (UniprotKB Q6ZMC9, residues 20–263) fused to mVENUS[57] after a TEV cleavage recognition site was synthesized, codon optimized for expression in human cells and cloned into pHLsec vector[58] between AgeI and KpnI restriction sites. The DNA encoding human Siglec-15$_{d1-d2}$ wild-type and R143A mutant, fused to the human IgG1 Fc region (UniprotKB P01857, residues 99–330) and with a C-terminal 6x His tag, was subcloned between XbaI and AfeI restriction sites into pcDNA 3.4 (Invitrogen) and codon optimized for expression in human cells. For the 5G12 Fab, the heavy (residues 1-113) and light (residues 1–107) chains were synthesized and cloned into pHLsec vector[58] between AgeI and KpnI restriction sites. All plasmids were synthesized by GenScript.

## Expression and purification of Siglec-15 and 5G12 Fab proteins

Siglec-15-mVENUS and Siglec-15-Fc (WT and R143A mutant) constructs were transiently transfected into HEK293F/S suspension cells. Cells were split in 200 mL cultures at $0.8 \times 10^6$ cells/mL. The DNA: FectoPRO transfection reagent solution (101000007, Polyplus) was then added directly to the cells, and cells were incubated at 37 °C, 130 rpm, 8% CO$_2$ and 70% humidity for 6–7 days. Supernatants were passed through a HisTrap Ni-NTA (17528601, GE Healthcare) and then separated on a Superdex 200 Increase size exclusion column (GE28-9909-44, GE Healthcare) in 20 mM Tris pH 9.0 (PHG0002, Sigma-Aldrich), 300 mM NaCl (S9888, Sigma-Aldrich) buffer to achieve size homogeneity. The heavy chain and light chain of 5G12 Fab were co-expressed at 2:1 ratio into HEK293F cells as described elsewhere[59]. The supernatant containing 5G12 Fab protein was flowed through a KappaSelect affinity (17545812, GE Healthcare) and eluted with 100 mM glycine pH 3.5. Eluted fractions were immediately neutralized with 1 M Tris-HCl pH 9.0. Fractions containing protein were pooled and run on a Superdex 200 Increase gel filtration column to obtain purified samples. Siglec-15–5G12 Fab complex was obtained by transiently co-transfecting Siglec-15-mVENUS with the heavy and light chains of 5G12 Fab into HEK293F suspension cells at 2:2:1 ratio. Expression was achieved following the same procedure described for Siglec-15-mVENUS alone. Supernatants were passed through a HisTrap Ni-NTA column (GE Healthcare). Siglec-15-5G12 protein complex was eluted with an increasing gradient of imidazole (up to 500 mM). Fractions containing Siglec-15-mVENUS–5G12 Fab complex were pooled and buffer exchanged to 20 mM Tris pH 9.0, 150 mM NaCl buffer to eliminate the imidazole. mVENUS protein was cleaved after incubation of 1 h at 37 °C with TEV enzyme at 20:1 molar ratio. TEV-treated sample was subsequently run on HisTrap Ni-NTA column (GE Healthcare). Siglec-15–5G12 Fab complex eluted from the column during 4 column volume wash with 20 mM Tris pH 9.0, 150 mM NaCl buffer and 500 mM Imidazole. The complex was concentrated and separated on a Superdex 200 Increase size exclusion column (GE Healthcare) in 20 mM Tris pH 9.0, 150 mM NaCl buffer to achieve size homogeneity.

## Crystallization, X-ray data collection and structure solution

Purified 5G12 Fab protein was concentrated to 10 mg/mL in a buffer containing 20 mM Tris pH 9.0 and 150 mM NaCl. Crystals were obtained by sitting drop vapor diffusion at 291 K in 20 % (w/v) PEG 3350, 0.1 M HEPES, pH 7.5 and 0.2 M MgCl$_2$ in 96-well plates after mixing 0.2 µL and 0.2 µL of protein and solution using Mosquito Crystal (SPT Labtech) crystallization robot. Crystals were cryo-protected by soaking them in mother liquor solution containing 25 % glycerol and flash cooled in liquid nitrogen. X-ray diffraction data was collected at the XALOC synchrotron beamline at ALBA (Spain). Data was processed using XDS in C121 space group at 3.9 Å resolution. The structure was determined by molecular replacement using the light chain and heavy chain of epratuzumab Fab (PDB ID 5VKK) as search model in Phaser.

Crystals of Siglec-15–5G12 Fab complex were obtained by hanging drop vapor diffusion at 291 K in 20 % (w/v) PEG 3350 and 0.2 M CaCl$_2$ in 24-well plates after mixing 1 µL of protein with 1 µL of solution. X-ray diffraction data was collected at the SLS synchrotron beamline at PXIII in Swiss Light Source (Switzerland). Data was processed using XDS in C121 space group at 2.1 Å resolution. The structure was determined by molecular replacement using the light chain and heavy chain of 5G12 Fab as model in Phaser[60]. The V Ig-like domain of Siglec-15 was manually built in Coot[61] and refined with Phenix[62] after several iterative rounds.

All structures were refined by manual building in Coot and using phenix.refine. PyMOL was utilized for structure analysis and figure rendering. All buried surface area values reported were calculated using EMBL-EBI PDBePISA. The crystal structures of 5G12 Fab and Siglec-15–5G12 Fab complex reported in this manuscript have been deposited in the Protein Data Bank, www.rcsb.org with PDB ID 7ZOZ and 7ZOR, respectively.

## Biolayer interferometry

The binding affinities of 5G12 Fab to Siglec-15-Fc was measured by BLI using the Octet R8 BLI system (Sartorius). Ni-NTA biosensors (18-5101, Sartorius) were hydrated in 1× kinetics buffer (PBS, pH 7.4, 0.002 % Tween, 0.01 % bovine serum albumin (BSA)) and loaded with 25 ng/µL of Siglec-15-Fc for 60 s at 1000 rpm. Biosensors were then transferred into wells containing 1× kinetics buffer to baseline for 60 s before being transferred into wells containing a serial dilution of Fab starting at 100 nM and decreasing to 6.25 nM. The 180 s association phase was subsequently followed by a 240 s dissociation step in 1× kinetics. Analysis was performed using the Octet software (Sartorius), with a 1:1 fit model. All experiments were repeated in triplicate, values were averaged, and standard errors were calculated.

## NMR experiments

All NMR experiments were recorded on a Bruker Avance III 600 MHz spectrometer equipped with a 5-mm inverse detection triple-resonance cryogenic probe head with z-gradients or in Bruker Avance III 800 MHz spectrometer equipped with a TCI cryoprobe. The 1-O-aminohexyl 3' sialyllactose (3'SL, OA32150), 1-O-aminohexyl 6'siallylactose (6'SL, OA32151) and STn-Ser (OA07388) were purchased from Carbosynth.

**NMR assignment.** $^1$H-NMR resonances of the ligands were assigned through standard 2D-TOCSY (30 ms mixing time), 2D-ROESY/NOESY (400 ms mixing time, respectively) and 2D $^1$H,$^{13}$C-HSQC experiments. The assignment was accomplished with ligands at concentrations ranging from 250 µM to 1.2 mM in 10 mM phosphate buffer (pH 7.5) with 300 mM NaCl in deuterated water (D$_2$O), at 298 K and 283 K. The resonance of 2,3-tetradeutero-3-trimethylsilylpropionic acid (TSP) was used as a chemical shift reference in the $^1$H-NMR experiments (δ TSP = 0 ppm).

**Saturation transfer difference (STD) NMR.** For STD-NMR experiments, Siglec-15-mVENUS (20 µM) in 10 mM phosphate buffer (pH 7.5) containing 300 mM NaCl and 0.05% sodium azide in D$_2$O. STD-NMR experiments were performed with a Siglec-15-mVENUS:3'SL, Siglec-15-mVENUS:6'SL and Siglec-15-mVENUS:STn-Ser at 1:40 molar ratios in 600 MHz at 283 K. The competition experiment with 5G12 Fab was performed with 20 µM of Siglec-15-mVENUS in presence of 30 µM of 5G12 Fab and 800 µM of 3'SL/6'SL/STn-Ser in 1:1.5:40 molar

ratio. Spectra were acquired with 1152 scans in a matrix with 64 K data points, in a spectral window of 12,335.5 Hz centered at 2818 Hz. An excitation sculpting module with gradients was used to suppress the water proton signals. Selective saturation of Siglec-15-mVENUS resonances (on resonance spectrum) was performed by irradiating at 7.2 ppm (aromatic residues) using a series of 40 Eburp2.1000-shaped 90° pulses (50 ms) for a total saturation time of 2 s, and a relaxation delay of 3 s. For the reference spectrum (off resonance), the samples were irradiated at 100 ppm. Control STD-NMR experiments were performed with ligands without Siglec-15-mVENUS and Siglec-15-mVENUS without ligands, at the same ligand and protein concentrations and using the same STD experimental setup. The STD spectra were obtained by subtracting the on-resonance spectrum to the off resonance spectrum. Then, the percentages of STD intensities were estimated by comparing the intensity of the signals in the STD spectrum with the signal intensities of the off resonance spectrum. The STD intensities of the ligands in absence of the protein and the residual STD intensities observed in the STD spectrum of Siglec-15-mVENUS were taken into account (subtracted) in the analysis of the STD spectrum of the complex. To determine the STD-derived epitope map of the ligands in presence of Siglec-15, the relative percentages of spin saturation of each proton were calculated by setting to 100 % the STD signal of the proton with the highest intensity and calculating the other STD signals accordingly. It was identified, on the STD epitopes, with asterisks (*), the resonances overlapped on the $^1$H-NMR spectrum.

The STD of Siglec-15-STn-CD11b/CD18 were acquired with Bruker 800 MHz spectrometer with a cryoprobe (Bruker, Billerica, MA, United States) at 298 K 30 μM of Siglec-15-mVENUS was mixed with 20 equivalents of STn, and then 0.05 eq of CD11b/CD18 was finally added. The on-resonance spectrum was performed by irradiating at 7 ppm (aromatic residues), with a saturation and relaxation time 2 s and 3 s, respectively.

**tr-NOESY experiments.** NOESY spectra of the ligands in absence and presence of Siglec-15-mVENUS were acquired on 600 (3′SL) or 800 MHz (6′SL and STn-Ser) spectrometer at 283 K. For the tr-NOESY samples with 450 μM of the ligands in presence of 30 μM of Siglec-15-mVENUS (Siglec-15/STn-Ser 1:15 molar ratio) in 10 mM phosphate buffer (pH 7.5) containing 300 mM NaCl and 0.05% sodium azide in D₂O were prepared. A sample with 450 μM of ligands in the absence of the protein but in the same buffer conditions were also prepared. 2D-NOESY spectra of the free ligands and Siglec-15/complexes (tr-NOESY) were acquired with 400 ms and with 150 ms of mixing time, respectively. The pulse program used is phase sensitive and suppresses the solvent signal with presaturation. Both NOESY spectra were acquired with 64 scans and 2048 × 256 (F2 × F1) points, with a spectral width of 9615.4 Hz centered at around 3760 Hz. The FID of each spectrum was Fourier-transformed with 2048 × 1024 (F2 × F1) points, and phase and baseline corrections were made.

**Molecular docking and dynamics simulations**
In the case of 3′SL and 6′SL derivatives, the crystal structure of Siglec-15 was superimposed with either mouse Sialoadhesin (PDB entry 1QFO)[63] or human Siglec-2 (PDB entry 5VKM)[64], allowing for the estimation of the coordinates of 3′SL or 6′SL, respectively, bound to Siglec-15. For STn-Ser, molecular docking calculations were performed with the AutoDock Vina program using the standard parameters (Supplementary Fig. 11). The torsion angles between αGalNAc and Ser were initially defined as φ2 = 66.3°, ψ2 = 179.5° and χ1 = 63.7°. For the selection of the docking solutions, the interaction between the carboxylate group of Neu5Ac and the guanidine side chain of R143 was a mandatory requisite. The coordinates of Siglec-15/3′SL, Siglec-15/6′SL complexes and the best pose in terms of binding energy derived from the docking calculations for STn-Ser were further subjected to MD simulations with

the AMBER package (v20)[65] using the force fields ff14SB[66] and GLYCAM06j-1[67]. The zwitterion in STn-Ser was generated with the antechamber module of AMBER with partial charges set to fit the electrostatic potential generated with HF/6-31G(d) by RESP[68] using Gaussian 16[69]. The complex was immersed in a 10 Å water box with TIP3P water molecules[70] and charge neutralized by adding explicit counter ions. A two-stage geometry optimization approach was carried out. The first stage minimizes only the positions of solvent molecules and ions, and the second stage is an unrestrained minimization of all the atoms in the simulation cell. The systems were then gently heated by incrementing the temperature from 0 to 300 K under a constant pressure of 1 atm and periodic boundary conditions. The time step was kept at 1 fs during the heating stages, allowing potential inhomogeneities to self-adjust. Water molecules are treated with the SHAKE algorithm. Long-range electrostatic effects are modeled using the particle-mesh-Ewald method[71]. An 8 Å cutoff was applied to Lennard-Jones and electrostatic interactions. Each system was equilibrated for 2 ns with a 2-fs time step at a constant volume and temperature of 300 K. Production trajectories were then run for additional 500 ns under the same simulation conditions.

**Flow cytometry**
For analysis of surface markers, T cells were collected at indicated time points after activation with CD3/CD28 Dynabeads (11131D, Thermo Fisher). For analysis of CD11b expression on T cells, activated T cells were collected at day 8, washed in Flow Cytometry Staining Buffer (00-4222-26, Thermo Fisher) and incubated with anti-CD11b biotin antibody (553309, BD Biosciences; 1:100) for 30 min at 4 °C. Cells where then washed and incubated with anti-CD3 BUV805 (612894, BD Biosciences; 1:100), anti-CD4 BUV395 (563550, BD Biosciences; 1:200) anti-CD8 APC/H7 (566855, BD Biosciences; 1:200) and streptavidin PE (12-4317-87, Thermo Fisher; 1:200) for 30 min at 4 °C in the dark. After washing, cells were resuspended in 200 μL staining buffer containing DAPI (1:10,000) (D1306, Invitrogen).

For STn surface staining, cells were incubated with anti-STn primary antibody (ab115957, Abcam; 1:100) for 30 min at 4 °C. Cells were then washed and incubated with an anti-mouse IgG-FITC secondary antibody (406001, Biolegend; 1:200) for 30 min at 4 °C in the dark. After a final wash step, cells were resuspended in 200 μL of 2 % BSA in PBS with DAPI (1:10,000). Data were collected on a FACSymphony flow cytometer and analyzed using FlowJo (BD Biosciences).

**CD11b in vitro blockade**
For blocking assays, T cells were preincubated with anti-human CD11b antibodies (clone M1/70, 557394, BD Biosciences or clone CBRM1/5, 14-0113-81, Thermo Fisher) or matched isotype controls (rIgG2a, 14-4321-82, Thermo Fisher and mIgG1, 14-4714-82, Thermo Fisher) at 10 μg/mL in 2 % BSA in PBS for 30 min at 4 °C. After washing, cells were incubated with recombinant Siglec-15-Fc, Siglec-15-Fc R143A mutant or human IgG1 Fc control (110-HG-100, R&D) at 4 μg/mL for 30 min at 4 °C. Cells were then washed and incubated with anti-human IgG Fc PE (12-4998-82, Thermo Fisher, 1:200), anti-CD4 BUV395 (563550, BD Biosciences, 1:200) and anti-CD8 BUV805 (612889, BD Biosciences 1:200) and incubated for 30 min at 4 °C in the dark. Cells were washed and resuspended in 200 μL 2 % BSA in PBS with DAPI (1:10,000) before acquisition on a FACSymphony.

**Lectin binding to T cells**
T cells (1 × 10⁶) were incubated with biotinylated SNA or MAL II (B-1305-2 and B-1265-1, Vectorlabs) at 5 μg/mL in 100 μL of 2% BSA in PBS for 1 h at 4 °C. After two washes at 400 g for 5 min, cells were incubated in 100 μL containing Streptavidin PE (1:200) (554061, BD Biosciences) and indicated fluorochrome-labeled antibodies for 20 min at 4 °C. Cells were then washed and resuspended in 200 μL of

2 % BSA in PBS with DAPI before acquisition in FACSymphony. Results were analyzed with FlowJo (BD Biosciences).

## Deglycosylation of T cells

Human T cells ($1 \times 10^6$) activated for 48 h were incubated with 0.3 mL of α2-3,6,8,9 Neuraminidase A (P0722, NEB) or Neuraminidase S (P0743L, NEB) at 37 °C for 1 h for removing sialic acid. For removing total glycosylation, activated T cells ($1 \times 10^6$) were treated with 8.5 μL of Protein Deglycosylation Mix II (P6044, NEB) and incubated at 37 °C for 30 min according to manufacturer's instructions. Deglycosylation efficiency was determined by lectin binding as described above.

## ELISA

ELISA plates were coated with 100 μL of recombinant CD11b/CD18 heterodimer (4047-AM-050; R&D Systems) at 2 μg/mL in carbonate-bicarbonate coating buffer overnight at 4 °C. Next day, plates were washed three times with PBST (PBS, 0.05% Tween-20, P2287, Sigma-Aldrich) and blocked with Carbo-free blocking solution (SP-5040-125, Vectorlabs) for 1 h at RT. Once removed the blocking buffer, 100 μL of indicated concentrations of Siglec-15-Fc WT/Siglec-15-Fc R143A mut/ IgG Fc recombinant proteins diluted in 1% Carbo-free blocking solution were added for 2 h RT. After three washing steps with 250 μL PBST, wells were incubated with anti-Fc HRP (A01854 200, Genscript) detection antibody diluted 1:5000 for 45 min at RT. Three washes of 250 μL with PBST, followed by another two washes with 400 μL of PBS were performed before the addition of 100 μL of TMB (34021, Thermo Fisher). The reaction was stopped with 50 μL of Stop Solution (N600, Thermo Fisher). Optical density (OD) was measured at 450 nm in a multimode plate reader (Victor Nivo, PerkinElmer).

## Immunoprecipitation

Sixty microliters of protein G sepharose magnetic beads (GE28-9440-08, Sigma-Aldrich) were washed in Buffer A (150 mM NaCl, 1 mM CaCl₂, 3 mM MnCl₂, 1 mM MgCl₂, 25 mM TRIS, 2 % BSA pH 7.5) and incubated with Siglec-15-Fc WT (10 μg/mL), Siglec-15-Fc R143A (10 μg/mL) or control IgG1-Fc (5 μg/mL) recombinant proteins in Buffer A containing protease inhibitors (ab287909, Abcam) for 1 h at 4 °C. After washing, coated beads were then incubated with 2.5 μg of recombinant CD11b/CD18 for 3 h in Buffer A. The bead complexes were then washed three times with buffer A and transferred to a clean tube. Samples were eluted at 95 °C for 5 min in reducing loading buffer and analyzed by SDS-PAGE. For FLAG-tagged CD11b precipitation, $30 \times 10^6$ transduced T cells were sonicated in RIPA buffer (89900, ThermoFisher). After a high speed centrifugation, the obtained protein extract was incubated with anti-DYKDDDDK magnetic agarose beads (A36797, Thermo-Fisher) overnight at 4 °C.

## Western blot

Total T cell lysates were collected using RIPA buffer (89900, Thermo-Fisher). Obtained protein extracts were incubated with PNGase F (2 μL, P0704L, NEB), Neuraminidase A (2 μL, P0722, NEB) or O-glycosidases (2 μL, P0733L, NEB) in 20 μL of total reaction volume of 1X Glycobuffer (B3704SVIAL, NEB) for 4 h at 37 °C. Immunoprecipitated and input samples were separated by 4–15 % Mini-PROTEAN TGX precast protein gel (4561083, BioRad) and transferred to a 0.2 μm PVDF membrane (1704156, BioRad) using a Trans-Blot Turbo transfer system (BioRad). The membrane was blocked for 1 h in 5% skim milk and 0.5 % Tween-20 diluted in PBS. An overnight incubation with primary antibodies (anti-CD11b, ab133357, Abcam) was performed, followed by five washes with PBS (containing 0.5% Tween-20) and incubation with secondary HRP-conjugated antibodies (1:5000). After the incubation with the secondary antibody, five additional washes were carried out and Chemi-luminescence detection was performed using Clarity Max Western ECL Substrate (170506, BioRad) on an iBright CL1500 system (Invitrogen).

For lectin-based western blot, the membrane was blocked with 1X Carbo-free blocking 1% Tween-20 solution (SP-5040-125, Vectorlabs) for 1 h at RT. Membranes were then incubated for an hour with 1 μg/mL SNA or MAL II in the same carbo-free blocking solution. After 3 washes with carbo-free solution, membranes were incubated with streptavidin-HRP at 1:5000 in the same buffer.

## Lentiviral plasmid construction

The coding sequences of human full-length CD11b and CD18 were synthetized and subcloned into a pLV-MSCV lentiviral vector (Genscript). A P2A-Blasticidin cassette and a P2A-Puromycin cassette were incorporated into CD11b pLV-MSCV and CD18 pLV-MSCV plasmids for selection purposes, respectively. For pulldown experiments, a C-terminal 3XFLAG tag was synthetized and inserted into CD11b pLV-MSCV vector (Genscript).

## Lentivirus production

To generate lentiviral particles, $5 \times 10^6$ 293T cells were seeded in a 100 mm dish. Twenty-four hours after cell seeding, cells were transfected using a mix of lentiviral plasmids (transfer plasmid; 5 μg psPAX2 (Addgene #12260; 4 μg) and VSV-G (Addgene #8454; 1.5 μg)) and jetPEI transfection reagent (101-10 N, Polyplus Transfection) following manufacturer's instructions. After 48 h, lentiviral particles were harvested from the supernatant, filtered through a 0.45 μm filter (514-0063, VWR) and concentrated using LentiX contentrator (631232, Takara) at a 3:1 ratio at 4 °C overnight. Lentiviral particles were then concentrated by centrifugation at $1500 \times g$ for 45 min, aliquoted and stored at −80 °C until use.

## Human T cell transduction

To express CD11b in human primary T cells using lentiral vectors, isolated CD3⁺ T cells were activated with Dynabeads in OpTimizer medium supplemented with IL-2. On day 2, $0.5 \times 10^6$ cells were transduced with CD11b and CD18 lentiviral particles at a multiplicity of infection (MOI) of 3 in the presence of Polybrene (TR-1003, Merck) at 8 μg/mL. Cells were spinoculated at $800 \times g$ for 1 h at 32 °C. After 48 h, puromycin and blasticidin were added for selection of double transduced population. Cells were assessed for transduction efficiency after 3-4 days by detection of CD11b surface expression by flow cytometry.

## siRNA CD11b knockdown

T cells were activated, debeaded and expanded as described before. At day 5, we isolated CD11b⁺ T cells by cell sorting (BD FACSAria Fusion) and further expanded in complete T cell media. At day 10, $0.25 \times 10^6$ T cells were transfected with Neon Transfection System (Thermo Fisher) using 500 nM of Silencer Select siRNA ITGAM (4392420) or negative control #1 (4390843) from Thermo Fisher. Electroporation was carried out using 10 μL NEON tips with the following parameters: 1600 V, 10 ms, 3 pulses. After transfection, cells were cultured for 48 h and CD11b downregulation was confirmed by flow cytometry.

## Proximity labeling

Activated T cell samples from two different donors were analyzed. Siglec-15-Fc WT or Siglec-15-Fc R143A was firstly preincubated with anti-hFc HRP (A01854 200, GenScript) for 30 min at 4 °C, forming the complex A, then mixed with T cells and incubated at 4 °C for 1 h. After washing steps, the labeling solution (TBS + 10 mM H₂O₂ + 95 μM Biotin Tyramide (LS-3500, Iris Biotech)) was added to samples and incubated for 7 min with shaking, and the reaction was stopped by adding the quenching buffer (TBS + 100 μM ascorbic acid). Samples were then incubated with 30 μL Protein A agarose nanobeads (29200, Thermo-fisher) for 60 min. The non-bound material was removed by washing the beads on microcentrifuge. The beads were eluted from the beads

using 300 μL labeling buffer (5 mM TCEP, 100 mM TRIS pH 8, 1% SDS, 0.1 mg/mL PMSF, Mammalian protease inhibitor, 0.1 M Sodium Thiocyanate). and an SDS-PAGE gel was performed.

## LC–MS/MS analysis

SDS-PAGE gel lanes were sliced into pieces as accurately as possible to guarantee reproducibility. The slices were subsequently washed in milli-Q water and reduction and alkylation were performed using dithiothreitol (10 mM DTT in 50 mM ammonium bicarbonate) at 56 °C for 20 min, followed by iodoacetamide (50 mM chloroacetamide in 50 mM ammonium bicarbonate) for another 20 min in the dark. Gel pieces were dried and incubated with trypsin (12.5 μg/mL in 50 mM ammonium bicarbonate) for 20 min on ice. After rehydration, the trypsin supernatant was discarded. Gel pieces were hydrated with 50 mM ammonium bicarbonate, and incubated overnight at 37 °C. After digestion, peptides were dried out in a RVC2 25 speedvac concentrator (Christ) and resuspended in 0.1% formic acid (FA). Peptides were further desalted, resuspended in 0.1% FA using C18 stage tips (Millipore), and sonicated for 5 min prior to analysis.

Samples were analyzed in a timsTOF Pro with PASEF (Bruker Daltonics) coupled online to an Evosep ONE liquid chromatograph (Evosep). 200 ng were directly loaded onto the Evosep ENDURANCE column (15 cm vs 150 μm, 1.9 μm) and resolved using the 30 samples-per-day standard protocol defined by the manufacturer (approximately 44 min runs). timsTOF mass spectrometer was operated in Data-Dependent Acquisition mode (DDA) using the standard 1.1. second acquisition cycle method (HyStar Version 5.1.8.1).

Protein identification and quantification was carried out using Byonic software (v2.16.11, Protein Metrics) through Proteome Discoverer v1.4 (Thermo Fisher). Searches were carried out against a database consisting of Homo sapiens (Uniprot/Swissprot, version 2020_04), with precursor and fragment tolerances of 20 ppm and 0.05 Da respectively. Carbamidomethylation of Cysteine was considered as fixed modification whereas oxidation of Methionine was considered as variable modification. A decoy search was carried out to estimate the false discovery rate (FDR) of the searches. Only proteins with at least one peptide identified at FDR < 1% were considered for further analysis. Spectral counts (SpC, the number of spectra that identifies peptides for a certain protein[72]) were used for the comparison of protein presence and abundance between conditions. Proteins with a SpC WT/Mut ratio>2, including those exclusively identified in the WT sample, were considered for further analysis and discussion.

The same digestion, acquisition and search protocol was applied to the analysis of the crystal samples that confirmed the presence of both d1 and d2 domains, except for that Mascot search engine (v2.2.07, Matrix Science) was used for the identification of the proteins.

### Quantification and statistical analysis

Statistical analyses were performed using GraphPad Prism version 8.0. The test applied in each panel is specified on the figure legends.

### Reporting summary

Further information on research design is available in the Nature Portfolio Reporting Summary linked to this article.

## Data availability

The crystalographic data of 5G12 Fab and Siglec-15−5G12 Fab complex generated in this study have been deposited in the Protein Data Bank database under accession codes 7ZOZ and 7ZOR. The mass spectrometry proteomics data have been deposited to the ProteomeXchange Consortium via the PRIDE partner repository with the dataset identifier PXD042009. Molecular dynamics simulations data, along with the top 10 poses from AutoDock Vina for Siglec15+STn-OMe, have been deposited in the "open science framework" repository and can be accessed at the following link: https://osf.io/ykgf5/?view_only= 21f8c01e396b456dadba577a842f49d9. Any remaining information can be obtained from the corresponding author upon request. Source data are provided with this paper.

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

## Acknowledgements

This work was supported by the European Research Council (ERC-2017-AdG, 788143-RECGLYCANMR to J.J.-B; ERC-2018-StG 804236-NEXT-GEN-IO to A.P.) and the Marie-Skłodowska-Curie actions (ITN Glytunes grant agreement No 956758 to K.S.; ITN BactiVax under grant agreement no. 860325 to U.A. and ITN DIRNANO grant agreement No 956544 to F.C.). X-ray diffraction experiments described in this paper were performed using beamlines XALOC synchrotron at ALBA (Spain) and PXIII in Swiss Light Source (Switzerland). F.M., C.S. and H.C. acknowledge Fundação para a Ciência e a Tecnologia (FCT-Portugal) for funding projects: PTDC/BIA-MIB/31028/2017 and UCIBIO project (UIDP/04378/2020 and UIDB/04378/2020) and Associate Laboratory Institute for Health and Bioeconomy—i4HB project (LA/P/0140/2020), to the CEEC contracts 2020.00233.CEECIND and 2020.03261.CEECIND for F.M. and H.C., respectively, and to PhD grant 2022.11723.BD of C.S. The NMR spectrometers are part of the National NMR Network (PTNMR) and are partially supported by Infrastructure Project No 22161 (co-financed by FEDER through COMPETE 2020, POCI and PORL and FCT through PID-DAC). F.M. and J.J.-B. acknowledge to the European funding for the GLYCOTwinning project (No. 101079417) and -COST Action GLYCONANOPROBES. A.P.'s research is funded by "La Caixa" Foundation (HR21-00925), AECC (LABAE211744PALA), Fundación FERO, Ikerbasque, and BIOEF EITB MARATOIA BIO19/CP/002. We thank Agencia Estatal de Investigación of Spain for grants PID2019-107956RA-I00 (A.P.), PID2019-107770RA-I00 (J.E.-O.), RTI2018-099592-B-C21 (F.C.), ID2020-114178GB (R.B. and J.D.S.), RYC2018-024183-I (A.P.), and the Severo Ochoa Center of Excellence Accreditation CEX2021-001136-S, all funded by MCIN/AEI/10.13039/501100011033 and by El FSE invierte en tu futuro, as well as CIBERES, and initiative of Instituto de Salud Carlos III (ISCIII, Spain) A.A.-V. receives funding from "La Caixa" Foundation (ID 100010434, LCF/BQ/DR20/11790022). A. B. (AECC Bizkaia Scientific Foundation, PRDVZ19003BOSC). F.C. acknowledges the Mizutani Foundation for Glycoscience (Grant 220115).

## Author contributions

Experimental conception and design: A.P., J.E.-O., J.J.-B.; data acquisition: M.P.L, L.E.M, A.A.V, J.I.Q, H.C., C.S., I.O., A. B., L.U., K.S., M.J.M., U.A., M.A., F.C., J.D.S., R.B., J.E.-O.; analysis of data: M.P.L., L.E.M., A.A.V., H.C., C.S., L.U., M.A., F.E., F.M., A.P., J.E.-O., J.J.-B.; drafting the article or revising it critically for important intellectual content: F.M., A.P., J.E.-O., J.J.-B.

## Competing interests

The authors declare no competing interests.

## Additional information

[1]Chemical Glycobiology lab, Center for Cooperative Research in Biosciences (CIC bioGUNE), Basque Research and Technology Alliance (BRTA), Bizkaia Technology Park, Building 800, 48160 Derio, Bizkaia, Spain. [2]Cancer Immunology and Immunotherapy Lab, Center for Cooperative Research in Biosciences (CIC bioGUNE), Basque Research and Technology Alliance (BRTA), Bizkaia Technology Park, Building 801A, 48160 Derio, Bizkaia, Spain. [3]Associate Laboratory i4HB—Institute for Health and Bioeconomy, NOVA School of Science and Technology, Caparica campus, 2829-516 Caparica, Portugal. [4]UCIBIO, Department of Chemistry, NOVA School of Science and Technology, Caparica campus, 2829-516 Caparica, Portugal. [5]Department of Chemistry, University of La Rioja, The Center for Research in Chemical Synthesis, Madre de Dios 53, E-26006 Logroño, Spain. [6]Ikerbasque, Basque Foundation for Science, Bilbao, Spain. [7]Proteomics Platform, CIC bioGUNE, CIBERehd, Basque Research and Technology Alliance (BRTA), Bizkaia Technology Park, Building 800, 48160 Derio, Spain. [8]Ubiquitin-likes and Development Lab, Center for Cooperative Research in Biosciences (CIC bioGUNE), Basque Research and Technology Alliance (BRTA), Bizkaia Technology Park, Building 801A, 48160 Derio, Bizkaia, Spain. [9]Department of Organic & Inorganic Chemistry, Faculty of Science and Technology, University of the Basque Country, EHU-UPV, 48940 Leioa, Bizkaia, Spain. [10]Centro de Investigacion Biomedica En Red de Enfermedades Respiratorias, 28029 Madrid, Spain. [11]These authors contributed equally: Maria Pia Lenza, Leire Egia-Mendikute, Asier Antoñana-Vildosola. ✉e-mail: jjbarbero@cicbiogune.es; apalazon@cicbiogune.es; jereno@cicbiogune.es

