## [Peer Review File · Nature Communications]

REVIEWER COMMENTS

Reviewer #1 (Remarks to the Author):

The submission from Ereno-Orbea et al is centered on Siglec-15, an important modulator of immune response to TAM; and that can be inhibited by specific mAb, this promoting antitumor responses. The authors solved the Crystallographic structure of the extracellular domain in complex with an antiSiglec-15 Ab, namely 5G12. Due to the flexibility of the other domains, only the V-domain gave the required electron density. The authors showed how the binding pocket is characterized by an extra b-strand, and the topology was similar to Sialoadhesin (Figure S3B can be moved to the main text). How many glycosylation sites are present in the V-domain?

The author used established NMR binding approaches to confirm sTn binds to Siglec-15 through the Sia portion. The authors in figure S8 (NOESY and trNOESY spectra) highlight two NOEs supporting the existence of tg and gt conformers around ω angle, but from Figure S9 it seems that during the MD simulation the tg rotamer was the only present, the authors should comment/justify this point, comparing experimental and MD results. The authors should run an MD simulation starting from the other rotamer. The authors should provide some more info on the docking analysis as a cluster analysis.

The binding of sTn is impaired by 5G12, as competition NMR experiments demonstrated. In this regards, an improvement of figure S10 and S5 could be useful; is the Ab glycosylated?

The ability of Siglec-15 to bind T-cell via sialylated glycans other than sTn, (although the structure of the recognized glycans has not been investigated) was also investigated, as also CD11b/CD18 as potential Siglec-15 binders. The manuscript uses a combination of established approaches and techniques to define 3D structure of V-set domain of Siglec-15 and the ligand binders. The experiments are well conducted. The topic is relevant. Particularly interesting are the structural insights into Siglec-5 binding site.

Reviewer #2 (Remarks to the Author):

Summary:

Siglec-15 is a member of the mammalian Siglec family, which binds to sialoglycans in a variety of immune functions. Many members of this family have been characterized both structurally and functionally, but the characterization of Siglec-15 is not as well developed as other Siglecs. this

importance of all the Siglecs is in recognition of self, which has broad implications for a range of diseases.

Here, the authors develop a Fab to stabilize Siglec-15 for in vitro and structural studies. They determine the structure by x-ray crystallography and show a minor modification of the V-set Ig fold. This is not the first time that modification of the “canonical” Ig folds has been reported. The authors use NMR analysis to investigate Ser-sialyl Tn antigen binding and suggest that this likely binds at the canonical sialic acid binding site. The authors then use immune cell lines and show that sialyl Tn antigen is not the physiological ligand on these cells but that one or more other unknown sialic acid-based ligands is likely the target. Finally, the authors show use proximity based assays and mass spec to suggest CD11b as one likely glycoprotein that is recognized by Siglec-15, validating this interaction by ELISA.

Major – scientific:

I suspect that some of these experiments were done when sTn was believed to be the primary ligand, then the manuscript was re-written following the discovery that Siglec-15 bound to cells without sTn. This has several implications for the paper. First, this makes things seem quite convoluted/confusing. In addition, it seems as if incorrect controls and comparators were used, and makes the manuscript seem incomplete. In short, the manuscript begins with a detailed molecular description of how Siglec-15 might interact Ser-sTn. Once it is shown that 2,3- or other 2,6-linked Sia-based glycans are likely the relevant ligands, the manuscript continues discuss work that focuses on sTn rather than pursuing and identifying binding properties of the ligands that are likely relevant. It is possible that Siglec-15 binds to multiple distinct Sia-based glycans. Thus, an advance might come in understanding how a single Siglec would support the binding of different ligands or by identifying one or more ‘true’ ligands and how the ligand preference might be determined biologically. This can be shown with competition assays or with purified proteins binding to purified ligands. Given that other 2,3- (presumably Sia-Gal) and 2,6-ligands are likely to be relevant, it may be worth the same level of detailed molecular description of the binding as is shown for sTn, which is not found on cells that Siglec-15 binds. Finally, it was never made clear what glycans are found on CD11b and which of these is likely to be recognized by Siglec-15. Because of this, the manuscript seems somewhat fragmented and not well integrated. Can the authors connect binding at the canonical binding site with a glycan that decorates T cells and CD11b? Can the authors revisit the NMR with an appropriate ligand? What is the relative binding of sTn and other potential sialoglycans? Do these ligands compete with each other and does Siglec-15 bind multiple glycans? How are the pieces of information in the results connected?

The variance on the isotype measurements shown in Figure S11 is high and this may require additional replicates to provide certainty in the measurement – this type of measurement should have a tighter grouping, as is observed in the anti-CD11b. This suggests a technical issue with a

subset of the measurements. Identifying the source of the technical issue and providing a more accurate measurement will allow confidence in whether this the difference between the two samples is statistically significant.

Minor - scientific:

Methods seem incomplete. The manuscript refers the reader to STAR methods (Cell press), which are not present. The source of sTn-Ser was not indicated – while common reagents need not necessarily have information about where they came from, this is an unusual reagent and its source ought to be specified. Was it synthesized?

Figure 3B – it is not clear what the “secondary control” is from reading the results and figure legend, and the legend does not describe the experiment well. How was this experiment performed? As a minor point, the key has a box color (off-white) that differs from the shaded peak (dark grey) that I think is supposed to be the secondary control. Figure 4 has lots of labeling problems that make this confusing – and this is exacerbated by use of truncated words like “de-sia” “unstim” “de-glyco”. Figure 4A has the same problem with the color designating the secondary control while Figure 4B has similar issues for “day 1”. Figure 4 x-axes – the arrow makes it appear as if the glycan is increasing in concentration from left to right.

Typo... the KD between Siglec-15 and the Fab is reported as 4.68 nM in the main text but as 46.68 nM in the table below Figure S1. Given that the error is ± 0.29 nM in each case, I'm guessing that the one in the SI has a bonus “6” before the decimal, but the authors are encouraged to check this discrepancy.

Major- non-scientific - presentation: this could use a read for both presentation and for grammar. Most grammar problems could probably be accomplished with an automated grammar checker such as from grammarly.com. However, it could be worth a read by a scientifically trained native speaker to ensure accuracy of wording selection and to remove colloquialisms and slang. Examples: Line 107 and throughout after “solved” the structure – is “determined” the structure meant? Line 135 and thereafter V-Ig domain (and sometimes V domain) – is a “domain organized around a V-set Ig fold” meant? If so, the V-set Ig fold should be explained to the readership. Line 139 and thereafter – what

is meant by d1 and d2 – is this how the lab refers to the domains of Siglecs? Note that the amount of copy editing is more than a reviewer would normally provide.

Minor:

It may be worth mentioning that “sialoadhesin” is also known as Siglec-1. As written, it seems that this is not part of the same family.

Lines 142-153: the detailed description of different strands of the Ig fold is difficult to follow.

Overuse of non-standard acronyms may make this challenging for even a reader in the field. Consider reducing the non-standard acronyms, particularly those that are not used frequently. BSA, HCDR, LCDR

Text that labels the main text figures is inconsistent in size/font/bolding and universally too small. The authors may want to consider something that is 1.5-2x the size of the text in Figure 1 and applying this throughout. Text that labels the supporting figures is inconsistent in size/font and generally too small to be useful. The authors may want to consider applying the font and size of figure labels in Figure S3 or Figures S7 to other supporting figures.

There are artifacts in some figures that may be from a layout program. For example, figure 1D has a box around the word ‘loop’. Figures 2A has a floating square outline in the lower left.

Figure 1 – the clarity could improve if there were boxes on the main figure that showed the location of the insets.

Figure 4B – the depiction of the SNFG symbols and the associated binding protein are of low resolution. The dark purple color of the SNFG sialic acid in the PDF differs from the standardized magenta enough to seem to be something different.

Figures S8 and S9C appear to be made at too low of a resolution and are pixelly and blurry. Consider re-rendering at higher resolution.

Reviewer #3 (Remarks to the Author):

Dr. Ereño-Orbea clearly shows that Siglec-15 binds to CD4+ and CD8+ T-cells in a sialic acid dependent manner, even though these cells did not express sialyl-Tn antigen. I only have two major comments on the nature of Siglec-15 ligand on T-cells and a few minor comments on the manuscript overall.

Major comments:

To further demonstrate Siglec-15 binds to α -2,3 linked sialoglycans on T-cells, T-cells could be treated with neuraminidase S, an α -2,3 specific neuraminidase, to show reduction of Siglec-15 binding to T-cells. Blocking lectin binding is not sufficient evidence to support this claim.

In Figure 5, Dr. Ereño-Orbea effectively show that Siglec-15 binds to recombinant CD11b/CD18 expressed in HEK293 using different methods but fail to show this interaction occurs in human T-cells from healthy subjects. Blocking binding with anti-CD11b antibody is not sufficient evidence that Siglec-15 directly interacts with CD11b. In order to further support the model in Figure 6, any one of the following experiments can be done:

1. Using human T-cell lysates, immunoprecipitate Siglec-15 natural ligand sialoglycan ligand and demonstrated CD11b and CD18 are eluted.
2. Knockdown CD11b in human T-cells and show reduced or abrogated binding of Siglec-15 to these cells.

Minor comments:

The sentence in lines 47-49 is not clear, I think the author meant to state: Siglec-15 binds to sialoglycan structures in T-cells that are not sialyl-Tn.

Dr. Ereño-Orbea is insufficiently clear on the statement in sentence describing Siglec-15 signaling cascade, lines 90-94. Siglec-15 contains a short cytoplasmic tail and no ITIM or ITSM motifs to my knowledge.

Dr. Ereño-Orbea mentions Siglec-15 binds to glycan structures, other than sialyl-Tn in paragraph starting in line 101, it is noteworthy to also mention Siglec-15 also binds sulfated sialylated glycan structures as shown independently by two different labs, references below:

1. Büll C, Nason R, Sun L, et al. Probing the binding specificities of human Siglecs by cell-based glycan arrays. *Proc Natl Acad Sci U S A*. 2021;118(17):e2026102118. doi:10.1073/pnas.2026102118

2. Jung J, Enterina JR, Bui DT, et al. Carbohydrate Sulfation As a Mechanism for Fine-Tuning Siglec Ligands. *ACS Chem Biol*. 2021;16(11):2673-2689. doi:10.1021/acscchembio.1c00501

The affinity of the 5G12 Fab KD reported in line 130 does not match the KD in supplementary Figure 1.

Dr. Ereño-Orbea does not provide enough methodological detail on how mass spectrometric analysis was done.

Suggest rewording the sentence in line 303

REVIEWER COMMENTS

Reviewer #1 (Remarks to the Author):

The submission from Ereno-Orbea et al is centered on Siglec-15, an important modulator of immune response to TAM; and that can be inhibited by specific mAb, this promoting antitumor responses. The authors solved the Crystallographic structure of the extracellular domain in complex with an antiSiglec-15 Ab, namely 5G12. Due to the flexibility of the other domains, only the V-domain gave the required electron density. The authors showed how the binding pocket is characterized by an extra b-strand, and the topology was similar to Sialoadhesis (Flguere S3B can be moved to the main text).

Many thanks. As suggested by the reviewer, we have moved the Supplementary Figure 3 (original manuscript) to Figure 1 as panel C and D.

How many glycosylation sites are present in the V-domain?

The V domain of Siglec-15 does not contain any N-linked glycans. The predicted N-linked glycan (N172) at the extracellular domain is located at d2. Therefore, we have included the following sentence in lines 143-144: "As expected, the electron density that justifies the presence of any N- or O-linked glycans on the surface of Siglec-15 was not observed."

The author used established NMR binding approaches to confirm sTn binds to Siglec-15 through the Sia portion. The authors in figure S8 (NOESY and trNOESY spectra) highlight two NOEs supporting the existence of tg and gt conformers around ω angle, but from Figure S9 it seems than during the MD simulation the tg rotamer was the only present0, the authors should comment/justify this point, comparing experimental and MD results. The authors should run an MD simulation starting from the other rotamer.

We thank the reviewer for pointing out this observation. This new version includes the full description of the molecular recognition of $\alpha(2,6)$ -sialyllactose (6'SL) and $\alpha(2,3)$ -sialyllactose (3'SL) ligands by Siglec-15 (see main and supplementary documents). In particular, the (NOESY and trNOESY) includes NMR data with 6'SL and 3'SL as ligands (Supplementary Figures 8 and 9), in addition to STn-Ser (Supplementary Figure 10). For all sialoglycans, the NOESY and trNOESY data were instrumental in the non-ambiguous determination of the φ dihedral angle in the free and bound states.

We have tried to deduce the preferences of ω in the free and bound states of 6'SL and STn-Ser from by analysing the relative NOE intensities between H5/H6proR; H5/H6proS; H4/H6proR; H4/H6proS in the NOESY and trNOESY spectra of these ligands. This analysis allowed us to exclude the gg conformer in the case of 6'SL but not to discriminate the relative population between gt and tg conformers of this ligand in either the free or bound states. In solution, $^3\text{J}_{\text{H5,H6ProR Gal}}$ and $^3\text{J}_{\text{H5,H6ProS Gal}}$ coupling constant analysis of 6'SL was performed and allowed to demonstrate that the major conformation in solution of

6'SL around ω angle corresponds to gt conformer (Supplementary Figure 9B). In the bound state, this analysis cannot be applied and based on the relative NOE intensity between H5/H6proR; H5/H6proS; H4/H6proR; H4/H6proS, we could not discriminate between gt and tg conformers (Supplementary Figure 9C). However, the presence of the gt conformer as major conformer in the bound state is also likely expected from the trend previously observed by other Siglecs (Ereño-Orbea et al. 2017; Forgione et al. 2020).

Unfortunately, the situation in the case of STn-Ser, is even more complicated due to the presence of conformers in Ser, either in absence (free state) or presence of Siglec-15 (bound state). An accurate analysis (NOESY free and bound state, coupling constant analysis in free state) to determine the orientation of ω angle was prevented by the strong overlap of various ¹H-NMR signals, including the H4, H5, H6proR and H6proS protons. In the first version of the manuscript, the caption of the Figure 2, which showed the H5/H6proR; H5/H6proS; H4/H6proR; H4/H6proS NOEs, incorrectly suggested the concomitant presence of these two conformers in the bound state. However, this was not the case, and we apologize for the misunderstanding. For STn-Ser we could not determine the orientation around ω angle. For this reason, in the case of STn-Ser, different MD simulations were carried out starting from gt and tg. The results of these simulations are now shown in Supplementary Figure 14. Particularly, we started with the gt conformer (60°), however, as the reviewer notes, the calculations show that after a certain time, only the tg conformer (180°) populates the trajectory. To explain this behaviour, we performed MD simulations with the ω dihedral fixed to 60° (gg conformation). The NMR analysis does not rule out the possibility of a gt conformer but restricting ω to a gt conformer leads to the loss of the CH- π interaction between Y87 and the C β of the serine residue, as well as the salt bridge between Arg157 and Ser (CO2-). The loss of these interactions may explain the preference of the calculations for the tg (180°) conformer, as indicated in the caption of the new figure. It is worth noting that MD simulations conducted on the complex of 6'SL and Siglec-15 exhibit a similar behaviour to that found for the Siglec-15/STn-Ser complex, with the tg conformer being the most populated (>90%, see Supplementary Figure 13), although a gt conformer of ω torsional angle was used in the initial structure. These results have been explained in the revised version of the manuscript.

The authors should provide some more info on the docking analysis as a cluster analysis.

It is worth noting that in the case of 3'SL and 6'SL bound to Siglec-15, the crystal structure of Siglec-15 was superimposed with either mouse Siglec-1 (PDB ID 1QFO) or human Siglec-2 (PDB ID 5VKM), allowing for the estimation of the coordinates of 3'SL or 6'SL, respectively. Docking analysis was performed to get an initial input structure for Siglec-15/STn-Ser complex. According to the referee's request, we have added a new figure (Supplementary Figure 11) in the supporting information. This figure includes the input file used in Vina, the complex with the best binding energy pose, the complex with the 20 lowest-energy poses suggested by vina, and a table with the pose, affinity, and RMSD values. These coordinates were used as starting structures in the MD simulations (Figure

4B in the manuscript and Supplementary Figures 12-14 in the revised version). The results of the new MD simulations and the experimental procedure has been discussed in the manuscript ("Molecular basis of of $\alpha(2,3)$ -, $\alpha(2,6)$ -sialyllactose and STn binding to Siglec-15 by NMR and molecular modelling" section of Results) and, in general, are in line with those derived from NMR experiments, being the Neu5Ac moiety the main binding contact. The new protocols have been included in "Molecular docking and dynamics simulations" section of Methods in the revised version of the manuscript.

The binding of sTn is impaired by 5G12, as competition NMR experiments demonstrated. In this regards, an improvement of figure S10 and S5 could be useful; is the Ab glycosylated?

With the objective of improving the description of the in-house production of 5G12 Ab Fab and to clarify that it is not glycosylated, we have modified the text (lines 121-126) "To assist the crystallization of the extracellular domain (ECD) of Siglec-15, that consists of two Ig domains (d1 and d2) (Siglec-15d1-d2), we employed the fragment antigen-binding (Fab) of an anti-Siglec-15 mAb as a crystallization chaperone (clone 5G1232). This non-glycosylated Fab was produced by cloning its variable heavy (VH) and light chains (VL) into a Fab scaffold containing human constant heavy (CH) and kappa light chains (CL)". We also have improved the Supplementary Figures 5 and 10, which are 4 and 15 in the new version, respectively.

The ability of Siglec-15 to bind T-cell via sialylated glycans other than sTn, (although the structure of the recognized glycans has not been investigated) was also investigated, as also CD11b/CD18 as potential Siglec-15 binders. The manuscript uses a combination of established approaches and techniques to define 3D structure of V-set domain of Siglec-15 and the ligand binders. The experiments are well conducted. The topic is relevant. Particularly interesting are the structural insights into Siglec-5 binding site.

We thank Reviewer 1 for the encouraging comments about this study. In this new version of the manuscript we have included the full description of the molecular recognition of $\alpha(2,6)$ -sialyllactose (6'SL) and $\alpha(2,3)$ -sialyllactose (3'SL) ligands by Siglec-15.

Reviewer #2 (Remarks to the Author):

Summary:

Siglec-15 is a member of the mammalian Siglec family, which binds to sialoglycans in a variety of immune functions. Many members of this family have been characterized both structurally and functionally, but the characterization of Siglec-15 is not as well developed as other Siglecs. this importance of all the Siglecs is in recognition of self, which has broad implications for a range of diseases.

Here, the authors develop a Fab to stabilize Siglec-15 for in vitro and structural studies.

They determine the structure by x-ray crystallography and show a minor modification of the V-set Ig fold. This is not the first time that modification of the “canonical” Ig folds has been reported. The authors use NMR analysis to investigate Ser-sialyl Tn antigen binding and suggest that this likely binds at the canonical sialic acid binding site. The authors then use immune cell lines and show that sialyl Tn antigen is not the physiological ligand on these cells but that one or more other unknown sialic acid-based ligands is likely the target. Finally, the authors show use proximity based assays and mass spec to suggest CD11b as one likely glycoprotein that is recognized by Siglec-15, validating this interaction by ELISA.

Major – scientific:

I suspect that some of these experiments were done when sTn was believed to be the primary ligand, then the manuscript was re-written following the discovery that Siglec-15 bound to cells without sTn. This has several implications for the paper. First, this makes things seem quite convoluted/confusing. In addition, it seems as if incorrect controls and comparators were used, and makes the manuscript seem incomplete. In short, the manuscript begins with a detailed molecular description of how Siglec-15 might interact Ser-sTn. Once it is shown that 2,3- or other 2,6-linked Sia-based glycans are likely the relevant ligands, the manuscript continues discuss work that focuses on sTn rather than pursuing and identifying binding properties of the ligands that are likely relevant. It is possible that Siglec-15 binds to multiple distinct Sia-based glycans. Thus, an advance might come in understanding how a single Siglec would support the binding of different ligands or by identifying one or more ‘true’ ligands and how the ligand preference might be determined biologically. This can be shown with competition assays or with purified proteins binding to purified ligands.

Thanks for pointing this out. We agree with the reviewer that other glycosylated ligands are possible. However, identifying additional sialylated ligands was beyond the scope of the current study. We have included the next sentence in the discussion (Lines 346-347) text: “However, we can not exclude that other glycoproteins may also contribute to the binding of Siglec-15 to T cells.”

Given that other 2,3- (presumably Sia-Gal) and 2,6-ligands are likely to be relevant, it may be worth the same level of detailed molecular description of the binding as is shown for sTn, which is not found on cells that Siglec-15 binds.

We thank the reviewer for the suggestions about clarity and manuscript organisation. We have reviewed the order of figures, the text, and added new data that directly address the raised concerns.

We have included data exploring the molecular basis of the interaction of Siglec-15 with α 2,6-Sia-Gal and α 2,3-Sia-Gal containing ligands, which was inspected using 1-O-aminohexyl 6’ sialyllactose (6’SL) and 1-O-aminohexyl 3’ sialyllactose (3’SL) derivatives. The corresponding STD-NMR and trNOESY binding data are extensively discussed in the

supporting information of the revised version of the manuscript (Figure 4, Supplementary Figures 6, 8-10). These results demonstrate that, for both ligands, the Neu5Ac moiety is the main contact point for the interaction with Siglec-15. 3D models of Siglec-15/3'SL and Siglec-15/6'SL were also generated, and their stability evaluated by 500 ns MD simulations. In these cases, the crystal structure of Siglec-15 was superimposed with either mouse Sialoadhesin (PDB ID 1QFO) or human Siglec-2 (PDB ID 5VKM), allowing for the estimation of the coordinates of 3'SL or 6'SL, respectively. These coordinates were found to be almost identical to those of the Neu5Ac moiety of the STn-Ser, as predicted by docking calculations, and we used them as initial structures in the new MD simulations (see new panels in Figure 4B and Supplementary Figures S10 and S11 in the revised version). The results of the new MD simulations are in line with those derived from NMR experiments, being the Neu5Ac moiety the main binding contact. These aspects and the new protocols have been discussed and included in the revised version.

Finally, it was never made clear what glycans are found on CD11b and which of these is likely to be recognized by Siglec-15. Because of this, the manuscript seems somewhat fragmented and not well integrated. Can the authors connect binding at the canonical binding site with a glycan that decorates T cells and CD11b?

Many thanks. We have included a new panel in Figure 6A showing that CD11b from T cells is highly N-glycosylated. Moreover, we have elucidated through SNA lectin blotting that CD11b from human T cells contains $\alpha(2,6)$ sialic acids (Figure 6B).

Can the authors revisit the NMR with an appropriate ligand?

We have determined the molecular basis for the interaction of 1-O-aminohexyl 3' sialyllactose (3'SL) and 1-O-aminohexyl 6' sialyllactose (6'SL) with Siglec-15 by STD-NMR and trNOESY experiments and results are included and discussed in the revised version of the manuscript (Figures 4, Supplementary Figures 6, 8-10).

What is the relative binding of sTn and other potential sialoglycans?

The affinities between sialylated mono, di or trisaccharides and Siglecs are rather low (low millimolar range). Thus, affinity measurements by biophysical techniques, such as isothermal titration calorimetry are out of range. We have included the following sentence in the Introduction (lines 99-101): "Binding constants of Siglecs for the N-acetylneuraminic acid (Neu5Ac) linked by $\alpha(2,3)$ - or $\alpha(2,6)$ - mono- or di-saccharides are in the low millimolar range (Kd of 0.1–3 mM) ^{37,38}."

Do these ligands compete with each other and does Siglec-15 bind multiple glycans?

STD-NMR competition binding experiments between 6'SL/3'SL and 5G12 for the Siglec-15 canonical binding site were carried out (Figure 3C). Similar conclusion as in the case of STn-Ser (Supplementary Figure 5) were observed using the 3'SL and 6'SL ligands and binding of 5G12 to Siglec-15 precludes these interactions.

The STD-NMR and trNOESY binding data obtained for 6'SL, 3'SL and STn with Siglec-15 show that, as with other Siglecs, binding mainly occurs through the Neu5Ac moiety. Thus, as shown by others (Murugesan et al. 2021), Siglec-15 can recognize with almost indistinction $\alpha(2,3)$ or $\alpha(2,6)$ sialylated ligands.

How are the pieces of information in the results connected?

We believe that the new data and re-organisation of the manuscript has improved the clarity, connection of results and readability of the manuscript.

The variance on the isotype measurements shown in Figure S11 is high and this may require additional replicates to provide certainty in the measurement – this type of measurement should have a tighter grouping, as is observed in the anti-CD11b. This suggests a technical issue with a subset of the measurements. Identifying the source of the technical issue and providing a more accurate measurement will allow confidence in whether this the difference between the two samples is statistically significant.

We have improved variability by repeating the experiment, obtaining similar results.

Minor - scientific:

Methods seem incomplete. The manuscript refers the reader to STAR methods (Cell press), which are not present.

We have updated the methods section accordingly.

The source of sTn-Ser was not indicated – while common reagents need not necessarily have information about where they came from, this is an unusual reagent and its source ought to be specified. Was it synthesized?

The 1-O-aminohexyl 3'sialyllactose (3'SL, OA32150), 1-O-aminohexyl 6'sialyllactose (6'SL, OA32151) and STn-Ser (OA07388) ligands were purchased from Carbosynth. We have included this information in Methods (lines 621-623).

Figure 3B – it is not clear what the "secondary control" is from reading the results and figure legend, and the legend does not describe the experiment well. How was this experiment performed? As a minor point, the key has a box color (off-white) that differs from the shaded peak (dark grey) that I think is supposed to be the secondary control.

To clarify the meaning of this control we have included the following sentence in the legend of Figure 3A: "Secondary control means that an anti-Fc detector antibody, but not recombinant Fc-chimera protein was added to the sample".

Figure 4 has lots of labeling problems that make this confusing – and this is exacerbated by use of truncated words like "de-sia" "unstim" "de-glyco". Figure 4A has the same problem with the color designating the secondary control while Figure 4B has similar

issues for "day 1". Figure 4 x-axes – the arrow makes it appear as if the glycan is increasing in concentration from left to right.

We thank the reviewer for these suggestions. Figure 4 (now Figure 2) has been edited for clarity including the figure legend. The secondary control was carried out by incubating cells with an anti-human IgG1-PE secondary antibody.

Typo... the KD between Siglec-15 and the Fab is reported as 4.68 nM in the main text but as 46.68 nM in the table below Figure S1. Given that the error is ± 0.29 nM in each case, I'm guessing that the one in the SI has a bonus "6" before the decimal, but the authors are encouraged to check this discrepancy.

We thank the Reviewer for pointing out this typo. We have corrected the incorrect value (46.68 nM) in the Table from Supplementary Figure 1, to 4.68 nM.

Major- non-scientific - presentation: this could use a read for both presentation and for grammar. Most grammar problems could probably be accomplished with an automated grammar checker such as from grammarly.com. However, it could be worth a read by a scientifically trained native speaker to ensure accuracy of wording selection and to remove colloquialisms and slang.

We have revised and edited the text. The new experiments were carried out in collaboration with Dr. Sutherland, a native English-speaker, who is now included as co-author. He has also assisted in this direction.

Examples: Line 107 and throughout after "solved" the structure – is "determined" the structure meant?

We have corrected this error as suggested by the reviewer in lines 45, 102, 119, 130, 587 and 595.

Line 135 and thereafter V-Ig domain (and sometimes V domain) – is a "domain organized around a V-set Ig fold" meant? If so, the V-set Ig fold should be explained to the readership.

We have hopefully improved the clarity of this section by changing and homogenising the terms to "V-set and C2-set Ig domains". Moreover, lines 79-83 read: "Siglec-15 presents an extracellular domain containing a conserved N-terminal variable (V)-set Ig domain, which binds sialic acid, and a constant 2 (C2)-set Ig domain 22. This V-set domain folds into a sandwich of two β -pleated sheets consisting of antiparallel β -strands and differs from C2-set by having additional β -strands within the β -sheets^{23,24}.

Line 139 and thereafter – what is meant by d1 and d2 – is this how the lab refers to the domains of Siglecs?

These acronyms have been clarified in the manuscript. For those Siglecs comprised of more than one C2-set Ig domains, we and others have used an immunoglobulin domain

numbering system to facilitate description of the 3D structures of the extracellular domains, ranging from Ig1-5 (as in MAG (Siglec-4) (Pronker et al. 2016)) or d1-d7 (as in CD22 (Ereño-Orbea et al, 2017 and 2021)).

Note that the amount of copy editing is more than a reviewer would normally provide.

We are grateful for the provided review.

Minor:

It may be worth mentioning that "sialoadhesin" is also known as Siglec-1. As written, it seems that this is not part of the same family.

We have corrected this point in line 153.

Lines 142-153: the detailed description of different strands of the Ig fold is difficult to follow.

We have moved Figure S3 to main Figure 1C and 1D, improving clarity and readability.

Overuse of non-standard acronyms may make this challenging for even a reader in the field. Consider reducing the non-standard acronyms, particularly those that are not used frequently. BSA, HCDR, LCDR.

The revised version of the manuscript includes a description of all acronyms.

Text that labels the main text figures is inconsistent in size/font/bolding and universally too small. The authors may want to consider something that is 1.5-2x the size of the text in Figure 1 and applying this throughout. Text that labels the supporting figures is inconsistent in size/font and generally too small to be useful. The authors may want to consider applying the font and size of figure labels in Figure S3 or Figures S7 to other supporting figures.

We have edited the text and figures to include the proposed changes.

There are artifacts in some figures that may be from a layout program. For example, figure 1D has a box around the word 'loop'. Figures 2A has a floating square outline in the lower left.

We have fixed these issues.

Figure 1 – the clarity could improve if there were boxes on the main figure that showed the location of the insets.

The location of the insets has been included in the Figure 1A.

Figure 4B – the depiction of the SNFG symbols and the associated binding protein are of low resolution. The dark purple color of the SNFG sialic acid in the PDF differs from the standardized magenta enough to seem to be something different.

The color of the SNFG symbols have been changed to standardized magenta and the resolution of the proteins was upgraded.

Figures S8 and S9C appear to be made at too low of a resolution and are pixelly and blurry. Consider re-rendering at higher resolution.

The image resolution has been improved, and high quality vector images (Illustrator Figures) will be provided for publication.

Reviewer #3 (Remarks to the Author):

Dr. Ereño-Orbea clearly shows that Siglec-15 binds to CD4+ and CD8+ T-cells in a sialic acid dependent manner, even though these cells did not express sialyl-Tn antigen. I only have two major comments on the nature of Siglec-15 ligand on T-cells and a few minor comments on the manuscript overall.

Major comments:

To further demonstrate Siglec-15 binds to α -2,3 linked sialoglycans on T-cells, T-cells could be treated with neuraminidase S, an α -2,3 specific neuraminidase, to show reduction of Siglec-15 binding to T-cells. Blocking lectin binding is not sufficient evidence to support this claim.

Yes, thanks for this suggestion. We have carried out the experiments proposed by the Reviewer, demonstrating that treatment of T cells with neuraminidase S results in a reduction of the binding of Siglec-15. The new results are now included in Figure 2G.

In Figure 5, Dr. Ereño-Orbea effectively show that Siglec-15 binds to recombinant CD11b/CD18 expressed in HEK293 using different methods but fail to show this interaction occurs in human T-cells from healthy subjects. Blocking binding with anti-CD11b antibody is not sufficient evidence that Siglec-15 directly interacts with CD11b. In order to further support the model in Figure 6, any one of the following experiments can be done:

1. Using human T-cell lysates, immunoprecipitate Siglec-15 natural ligand sialoglycan ligand and demonstrated CD11b and CD18 are eluted.
2. Knockdown CD11b in human T-cells and show reduced or abrogated binding of Siglec-15 to these cells.

In order to address this relevant concern, we have carried out additional cellular binding assays resulting in the addition of four new panels in Figure 5. First, we reduced the levels of surface CD11b expression on T cells by siRNA electroporation, which resulted in a decrease of the

binding of Siglec-15 to both human CD4+ and CD8+ T cells. In addition, we overexpressed CD11b/CD18 in T cells by lentiviral infection, which resulted in an increase of the binding of Siglec-15 to T cells. These new data complement previous experiments performed with an anti-CD11b blocking antibody (clone M1/70). These new results have been included and explained in lines 283-287.

Minor comments:

The sentence in lines 47-49 is not clear, I think the author meant to state: Siglec-15 binds to sialoglycan structures in T-cells that are not sialyl-Tn.

The reviewer is correct. This sentence has been clarified (lines 50-51): "Binding of Siglec-15 to T cells, which lack STn expression, depends on the presence of $\alpha(2,3)$ - and $\alpha(2,6)$ - linked sialoglycans."

Dr. Ereño-Orbea is insufficiently clear on the statement in sentence describing Siglec-15 signaling cascade, lines 90-94. Siglec-15 contains a short cytoplasmic tail and no ITIM or ITSM motifs to my knowledge.

We thank the reviewer for pointing this out. Human Siglec-15 has a transmembrane domain that contains a lysine residue (Lys274) that interacts with adapter protein DAP12, and a cytoplasmic tail (Ishida-Kitagawa., 2012). We have modified lines 83-86 in the main manuscript.

Dr. Ereño-Orbea mentions Siglec-15 binds to glycan structures, other than sialyl-Tn in paragraph starting in line 101, it is noteworthy to also mention Siglec-15 also binds sulfated sialylated glycan structures as shown independently by two different labs, references below:

1. Büll C, Nason R, Sun L, et al. Probing the binding specificities of human Siglecs by cell-based glycan arrays. Proc Natl Acad Sci U S A. 2021;118(17):e2026102118.

doi:10.1073/pnas.2026102118

2. Jung J, Enterina JR, Bui DT, et al. Carbohydrate Sulfation As a Mechanism for Fine-Tuning Siglec Ligands. ACS Chem Biol. 2021;16(11):2673-2689.

doi:10.1021/acscchembio.1c00501

We thank the reviewer for this suggestion. We have included the next sentence, and the suggested references, in lines 98-99: Moreover, Siglec-15 shows robust binding to sulfated sialic acid containing-glycans^{35,36}."

The affinity of the 5G12 Fab KD reported in line 130 does not match the KD in supplementary Figure 1.

We thank the reviewer for pointing this out, we have corrected the typo error in Supplementary Figure 1.

Dr. Ereño-Orbea does not provide enough methodological detail on how mass spectrometric analysis was done.

We have included more details at the methodological section entitled "LC-MS/MS analysis".

Suggest rewording the sentence in line 303.

This sentence has been rephrased now in lines 311-313: "Moreover, the presence of $\alpha(2,3)$ and $\alpha(2,6)$ sialylated glycans in T cells as the main binders for Siglec-15 has also been shown".

REVIEWERS' COMMENTS

Reviewer #1 (Remarks to the Author):

The authors have well addressed points and comments I raised, they also add new data and add (or improved) new figures. The manuscript in my opinion can be published.

Reviewer #2 (Remarks to the Author):

concerns were addressed. the revised manuscript is nicely improved.

Reviewer #3 (Remarks to the Author):

Dr. Ereño-Orbea has demonstrated Siglec-15 binds to CD4+ and CD8+ T-cells in a sialic acid dependent manner. Treatment of T-cells with neuraminidase S reduces binding, congruent with the author's conclusion that Siglec-15 binds to α -2,3 and α -2,6 linked sialoglycans on T-cells. The author also shows that reducing the surface expression of CD11b, results in reduced binding of Siglec-15 supporting their conclusion that CD11b carriers sialoglycan ligands that Siglec-15 binds to, further supported by their overexpression data. The author's conclusions are supported by the data and provides new insights into the molecules that Siglec-15 interacts with on T-cells.